# *Bad-PFL*: Exploring Backdoor Attacks against Personalized Federated Learning

**Mingyuan Fan**[1], **Zhanyi Hu**[1], **Fuyi Wang**[2], **Cen Chen**[1]*
[1]East China Normal Unversity,  [2]RMIT University
`fmy2660966@gmail.com`
`51255903110@stu.ecnu.edu.cn`
`fuyi.wang@rmit.edu.au`
`cenchen@dase.ecnu.edu.cn`

## Abstract

Data heterogeneity and backdoor attacks rank among the most significant challenges facing federated learning (FL). For data heterogeneity, personalized federated learning (PFL) enables each client to maintain a private personalized model to cater to client-specific knowledge. Meanwhile, vanilla FL has proven vulnerable to backdoor attacks. However, recent advancements in PFL community have demonstrated a potential immunity against such attacks. This paper explores this intersection further, revealing that existing federated backdoor attacks fail in PFL because backdoors about manually designed triggers struggle to survive in personalized models. To tackle this, we design *Bad-PFL*, which employs features from natural data as our trigger. As long as the model is trained on natural data, it inevitably embeds the backdoor associated with our trigger, ensuring its longevity in personalized models. Moreover, our trigger undergoes mutual reinforcement training with the model, further solidifying the backdoor's durability and enhancing attack effectiveness. The large-scale experiments across three benchmark datasets demonstrate the superior performance of *Bad-PFL* against various PFL methods, even when equipped with state-of-the-art defense mechanisms. The source codes are available in `https://github.com/fmy266/Bad-PFL`.

## 1 Introduction

Federated learning (FL) (Ye et al., 2024a; Fan et al., 2023) has become the de facto privacy-preserving framework for training neural networks. In FL, a server maintains a globally shared model and broadcasts it to selected clients. The clients train the model on their private datasets for several local steps and then return their models to be aggregated into a new global model, all without sharing sensitive information with the server or across devices. However, the server is limited in verifying clients' integrity, as doing so may infringe on privacy or fairness constraints (Ye et al., 2024a). This limitation opens the door for potential backdoor attacks in FL (Bagdasaryan et al., 2020; Zhang et al., 2022), where an adversary can compromise specific clients to upload backdoored models, so as to inject a backdoor into the global model. Once a predefined trigger is injected into the input data, the backdoor is activated, causing the model to misclassify the data into attacker-chosen labels.

Another challenge in FL is data heterogeneity across different clients, known as the non-IID problem (Ye et al., 2024a; Qin et al., 2023). The problem not only slows down the convergence of the global model but can also limit the model's performance for some clients. To tackle this, personalized federated learning (PFL) has emerged as a promising solution, learning a personalized model for each client to better fit their specific data distribution. At a high level, most PFL methods can be categorized into two groups: full model-sharing (Chen et al., 2022; Karimireddy et al., 2020) and partial model-sharing methods (Li et al., 2021b; Collins et al., 2021). In full model-sharing methods, clients adapt the global model using techniques such as fine-tuning with local datasets (Chen et al., 2022), parallel training of global and local models (Li et al., 2021a), regularization of the global model (Li et al., 2020), etc. In contrast, partial model-sharing methods share only specific parts of

---

*Corresponding Author.

the model. For example, FedRep (Collins et al., 2021) synchronizes the feature extractor of both the global and local models while maintaining each client's private classification head, and FedBN (Li et al., 2021b) shares all model parameters except for batch normalization layers.

Since the non-IID problem is ubiquitous in real-world environments (Ye et al., 2024a), it is particularly important to explore the robustness of PFL against backdoor attacks. Preliminary studies (Qin et al., 2023) have shown a positive conclusion: PFL not only mitigates the non-IID problem but also offers immunity to backdoor attacks. Specifically, partial model-sharing methods demonstrate significant resistance to existing federated backdoor attacks, reducing the attack success rate (ASR) to below 30%. The dual benefits make PFL quite popular and significantly alleviate users' concerns about the vulnerability of FL to backdoor attacks. However, upon reviewing the current literature on federated backdoor attacks (Xie et al., 2020; Liu et al., 2024; Zhuang et al., 2024), we find that existing attacks seem to inadequately exploit the characteristics of PFL, making the robustness of PFL against such attacks questionable. While some studies (Lai et al., 2022; Lyu et al., 2024) have investigated backdoor attacks PFL methods, they often concentrate too narrowly on specific approaches or can be easily mitigated by defense mechanisms. In light of this, we first explore to address the question: ***What causes existing federated backdoor attacks to fail in PFL?***

**Our contribution.** We identify three key factors contributing to the failure of these attacks. First, we empirically demonstrate that globally shared models can often be backdoored. However, in PFL, the personalized model typically differs from the global model or shares only partial parameters. In the former case, despite the presence of regularization terms that align the personalized model with the global model, the backdoor in the global model cannot be effectively implanted into the personalized model. Secondly, when only partial parameters are shared, the backdoor in the personalized model remains dormant, as the unshared private parameters are not adapted to accommodate the backdoor. Finally, since the personalized model is trained on clean data, the backdoor is gradually diluted due to catastrophic forgetting, further undermining attack effectiveness.

We develop a novel yet highly effective backdoor attack method, called *Bad-PFL*. Unlike existing federated backdoor attack methods that force models to learn predefined triggers, *Bad-PFL* exploits inherent backdoors within both the global and personalized models themselves. These backdoors correspond to the natural features of the attacker-chosen labels. We employ a generator to identify these features that make given samples appear most similar to the target category when viewed by the model but remain imperceptible to human observers. Moreover, we introduce disruptive noise that eliminates features associated with the ground-truth labels in the given data, so as to enable the natural features of the attacker-chosen labels to stand out during the model's decision-making process. The blend of these two types of noise forms the trigger of *Bad-PFL*, which is capable of effectively tricking the model into mistaking trigger-added data. The generator alternates optimization with the global model, further enhancing the trigger's effectiveness through mutual adaptation. Furthermore, due to the similarity between the global model and personalized models (regularization constraints, learned similar underlying features, etc.), the trigger can also effectively activate hidden natural backdoors within personalized models. These backdoors persist as long as personalized models are trained on natural data. By the way, the trigger is sample-specific and invisible, rendering *Bad-PFL* highly stealthy and difficult to detect (see Appendix B.2.11).

## 2    A CLOSE LOOK AT BACKDOOR ATTACKS IN PFL

### 2.1    FL VERSUS PFL

Consider a FL setup with $m$ clients and a central server. Let $\mathcal{U} = \{1, 2, \cdots, m\}$. Each client $i \in \mathcal{U}$ possesses a private dataset $D_i$ drawn from its local data distribution $P_{XY}^i$ defined over $\mathcal{X} \times \mathcal{Y}$, where $\mathcal{X}$ is the input space and $\mathcal{Y}$ is the label space with $K$ categories. Let $\mathcal{L} : \mathcal{X} \times \mathcal{Y} \to \mathbb{R}_+$ denote the loss function, e.g., cross-entropy loss. FL formulates the following optimization objective to train a global model $F$ parameterized by $\theta_g$:

$$\min_{w_g} \sum_{i \in \mathcal{U}} \mathbb{E}_{(x,y) \sim D_i} \left[ \mathcal{L}(F(x; \theta_g), y) \right]. \tag{1}$$

where $i$-th term is the local optimization objective for $i$-th client. FedAvg (McMahan et al., 2017) addresses Equation 1 by iteratively alternating between two steps: 1) participating clients download

the current global model $F(\cdot; \theta_g)$ from the server and then train the model on their respective local datasets, and 2) the server averages the locally trained models to form the new global model for the next iteration. However, it is common that $P_{XY}^i$ and $P_{XY}^j$ ($i \neq j$) are different (i.e., non-IID problem), potentially causing the global model to perform poorly for some clients. To mitigate the non-IID problem, PFL allows each client to maintain a private personalized model locally. For full model-sharing methods (Li et al., 2021a; T Dinh et al., 2020), Equation 1 is adjusted as follows:

$$\min_{\theta_i, i \in \mathcal{U}} \sum_{i \in \mathcal{U}} \mathbb{E}_{(x,y) \sim D_i} [\mathcal{L}(F(x; \theta_i), y) + \mathcal{R}(\theta_i, \theta_g)], \tag{2}$$

where $\theta_i$ is the parameters of $i$-th client' personalized model, and $\mathcal{R}$ serves as a regularization term that governs the distance between $\theta_i$ and $\theta_g$. In Equation 2, $\theta_i$ can be harnessed to learn client-specific knowledge since $\theta_i$ are not required to be identical to $\theta_g$. At the same time, $\mathcal{R}$ facilitates the integration of global knowledge into the personalized models.

For partial model-sharing methods (Li et al., 2021b; Xu et al., 2023), most elements in $\theta_i$ are constrained to be same as the corresponding elements in $\theta_g$, with only a few allowed to update freely. Let $\Lambda$ denote a set containing the indices where partial model-sharing methods enforce equality between $\theta_i$ and $\theta_g$ totally. We can summarize the partial model-sharing methods as follows:

$$\min_{\theta_i, i \in \mathcal{U}} \sum_{i \in \mathcal{U}} \mathbb{E}_{(x,y) \sim D_i} [\mathcal{L}(F(x; \theta_i), y)], \; s.t., \; \theta_g[k] = \theta_i[k], \; k \in \Lambda, \; i \in \mathcal{U}. \tag{3}$$

Compared to full model-sharing methods, partial model-sharing methods generally yield better performance (Xu et al., 2023), because they incorporate prior knowledge to identify which parameters are more likely to contain global knowledge and thus should be shared across clients, while allowing others that are more likely to learn client-specific knowledge to vary freely. For instance, in FedRep (Collins et al., 2021), the parameters of the feature extraction layer are shared across clients and only classification heads remain private. To avoid ambiguity, the local model refers to the global model that clients download from the server, excluding personalized models. In practice, to address Equation 2 and Equation 3, PFL methods may train $\theta_g$ locally according to Equation 1 before optimizing the personalized model (using the updated $\theta_g$), or vice versa.

## 2.2 THE CHALLENGES OF BACKDOOR ATTACKS IN PFL

Federated backdoor attacks consider an adversary that can manipulate certain clients to pollute the global model by inserting a backdoor task into the training process of their local models as follows:

$$\mathbb{E}_{(x,y) \sim D_i} [(1 - \alpha) \cdot \mathcal{L}(F(x; \theta_g), y) + \alpha \cdot \mathcal{L}(F(x + \mathcal{T}(x); \theta_g), y_t)], \tag{4}$$

where $\mathcal{T}$ is the trigger generation function, $y_t$ is the target label, and $\alpha$ is referred to as the poisoning rate, indicating the importance of the backdoor task relative to the main task. In the IID setting, the solution to Equation 4, i.e., the local solution of the compromised clients, can also perform well on datasets from other clients (Bagdasaryan et al., 2020). As a result, the global model ultimately converges to the solution of Equation 4, thereby embedding the backdoor into the global model. However, in the non-IID setting, a client's local solution may perform poorly on datasets from other clients (Ye et al., 2024a), suggesting that the global model may not necessarily converge to the solution of Equation 4. This issue is even more pronounced in PFL, where backdoor attacks become more challenging because each client's personalized model diverges from the global model. Concretely, we identify three key challenges that hinder backdoor insert in PFL.

***In full model-sharing methods, the regularization term alone is insufficient to directly transfer the backdoor from the global model to personalized models.*** We assess the performance of two backdoor attacks, Neurotoxin (Zhang et al., 2022) and PGD-Bkd (Wang et al., 2020a), across three full model-sharing methods: FedProx (Li et al., 2020), Ditto (Li et al., 2021a), and pFedMe (T Dinh et al., 2020). The detailed experimental settings and results for this section all can be found in Appendix B.1. In the non-IID setting, the ASR for the global model reaches approximately 90%. However, for personalized models, the ASRs typically range from 60% to 80%. This indicates that while the regularization term somewhat facilitates transferring the backdoor to personalized models, it is not entirely sufficient. To further validate this, we investigate the impact of removing the regularization term, which leads to a significant decrease in ASRs for personalized models, dropping to just 10%. Finally, we assess the gradients of the global model parameters in the presence

of trigger-embedded data. By weighting the regularization term based on the gradient magnitudes, the ASRs for personalized models rebound to levels close to the ASRs for the global model.

***In partial model-sharing methods, non-shared parameters obstruct the backdoor effect, making it difficult for the backdoor to influence the decision-making process of personalized models.*** We evaluate the robustness of two partial model-sharing methods, FedBN (Li et al., 2021b) and FedRep (Collins et al., 2021), against the two attacks mentioned above. Appendix B.1 reports the ASRs with three configurations. The first configuration purely applies three backdoor attack methods to both FedBN and FedRep. The second configuration fixes the shared parameters and fine-tunes only the non-shared parameters on the trigger-embedded data $((x + \mathcal{T}(x), y_t))$ and clean data for one epoch (15 steps). In this configuration, the ASRs recover to nearly 100%, as the non-shared parameters adjust to accommodate the backdoor. To exclude the possibility that the non-shared parameters are embedded with the backdoor during fine-tuning, we introduce a third configuration without any backdoor attacks. This guarantees that the trained personalized models remain free of backdoors. We fine-tune these clean models similarly to the second configuration and see that the ASRs for both FedBN and FedRep stay around 10%. This demonstrates that while the backdoor is indeed embedded within the shared parameters, the non-shared parameters do not adapt to the backdoor, thereby limiting the backdoor's effectiveness against partial model-sharing methods.

***During the training process, the backdoor could be progressively diluted.*** There are two primary scenarios where this dilution occurs. ***1)*** If compromised clients are picked by the server with a long gap, the backdoor within the global model may be gradually erased due to updates from benign models. This can further affect the embedding of the backdoor into personalized models. Notably, this scenario arises only in the non-IID setting. In the IID setting, the backdoored model from a compromised client can also be well-fitted to data from other clients. As a result, even if benign clients train the backdoored model locally, its parameters remain largely unchanged. Once the server replaces the global model with the backdoored one, the backdoor will persist (Bagdasaryan et al., 2020). ***2)*** After the completion of FL, clients may further fine-tune their personalized models using local datasets (Chen et al., 2022). This fine-tuning can reduce the backdoor's effectiveness, as the local data of benign clients does not contain the backdoor's triggers and thus could help to overwrite the embedded backdoor. Appendix B.1 validates these, demonstrating that in the non-IID setting, as the expected time for selecting malicious clients increases, the ASR for the global model declines. In IID settings, this decline is less pronounced. Similarly, when clients fine-tune their personalized models with clean data, the ASRs for these personalized models also gradually decrease.

**Motivation.** While we identify several measures that could potentially circumvent the robustness of PFL against backdoor attacks, these measures are often impractical. They require the adversary to manipulate the local training process of benign clients, such as using weighted regularization or contaminating their local datasets. Therefore, it is necessary to explore whether a practical federated backdoor attack method exists that can overcome these challenges.

## 3 OUR ATTACK: *Bad-PFL*

### 3.1 THREAT MODEL

**Adversary's objective.** We consider a practical attack scenario where the adversary compromises several clients to inject a backdoor into personalized models. The adversary expects these models to exhibit predetermined misclassification behavior for trigger-containing data while performing well on clean data. Moreover, the adversary desires that the embedded backdoors within these models are hard to detect or remove, so as to ensure the backdoors' stealthiness and longevity.

**Adversary's knowledge and capability.** The adversary has complete control over the compromised clients. This scenario is quite realistic, as each client has full sovereignty over their local training process and external entities cannot scrutinize their actions. Any client may launch a backdoor attack for personal gain or collaborate with others to execute one. Besides, to maintain the general applicability of *Bad-PFL*, the adversary lacks the ability to control the server's privileges (e.g., aggregation rule and client selection) or interfere with the training processes of benign clients.

---

**Algorithm 1:** PFL process with *Bad-PFL*

---

1 **Server Executes:**
2     Initialize the parameters of the global model $\theta_g$ ;
3     **while** *the global model is not converged* **do**
4         Broadcast the parameters $\theta_g$ to selected clients for local training (**ClientUpdate**);
5         Aggregate the parameters uploaded by selected clients to form a new global model;
6     **end**
7 **ClientUpdate:**
8     **if** *this client is compromised* **then**
9         Download $\mathcal{G}_w$ from the adversary and train it based on Equation 7 ;
10         Return trained $\mathcal{G}_w$ to the adversary ;
11         Train $F(\cdot\,;\theta_g)$ on its private dataset based on Equation 4 ;
12     **else**
13         Train $F(\cdot\,;\theta_g)$ on its private dataset based on Equation 4 with $\alpha = 0$ ;
14     **end**
15     Train the personalized model with the predefined PFL method ;
16     Return the new $\theta_g$ to the server ;

---

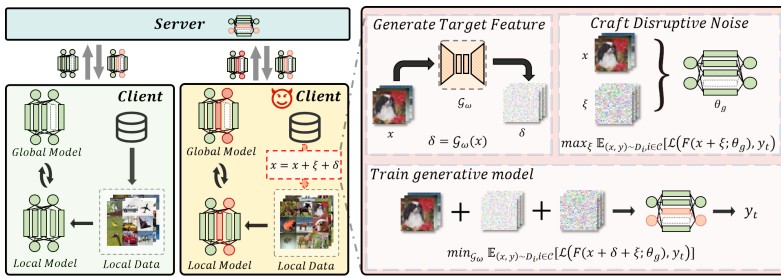

Figure 1: The overview of *Bad-PFL*. The right is the trigger generation function of *Bad-PFL*. When a compromised client is selected, it first trains the generator, and then uses this function to add triggers to the clean data for training its local model. Following this, the client trains its personalized model and uploads the backdoored local model.

### 3.2 *Bad-PFL*

**Overview.** As illustrated in Figure 1 and Algorithm 1, the attack process of *Bad-PFL* is similar to that of conventional federated backdoor attacks (Zhang et al., 2022; Zhuang et al., 2024). When the server selects a compromised client, the client optimizes Equation 4 and uploads the backdoored model to the server for aggregation. The key distinction of *Bad-PFL* lies in its trigger generation function $\mathcal{T}$, which combines target features with disruptive noise to create the trigger. We begin by presenting the intuition behind our trigger generation function.

**Intuition.** The core idea is to utilize the features of the target label as the trigger. Many clients' datasets contain samples of the target label, which can cause benign clients' personalized models to inadvertently learn the relationship between these features and the target label, stealthily embedding the backdoor into their personalized models. Even if some clients have datasets with few or no samples of the target label, the global model is likely to learn this relationship and then transmit it to personalized models. In contrast, existing backdoor attacks (Xie et al., 2020; Liu et al., 2024; Zhuang et al., 2024) rely on manually crafted features as triggers, which often do not exist in natural data. As demonstrated in Section 2, the backdoors associated with manually crafted triggers are often difficult to implant into personalized models.

**Trigger generation function.** Our trigger generation function consists of two key steps. The goal of the first step is to generate natural features $\delta$ associated with the target label $y_t$. Since these features mimic the patterns of the target label, the model will mistakenly believe that the data with these features belongs to the target label. However, these features $\delta$ may not completely dominate the input's characteristics, allowing for potential correct classification into the ground-truth label. To

handle this, the second step aims to design a noise $\xi$ that disrupts the features associated with ground-truth label $y$. By introducing this disruptive noise $\xi$, the model will rely more on $\delta$ for prediction as the features associated with ground-truth labels are corrupted. To summarize, our trigger generation function is defined as $\mathcal{T}(x) = x + \delta + \xi$ where $\delta + \xi$ serves as the trigger for $x$.

**Generate target feature $\delta$.** The primary challenge here lies in identifying the natural features of the target label. To this end, we can initialize a random noise $\delta$ and solve the following optimization task to uncover the target label's features:

$$\delta = \arg\min_{\delta} \mathbb{E}_{(x,y)\sim D_i, i\in\mathcal{C}} \left[\mathcal{L}(F(x + \delta; \theta_g), y_t)\right], s.t., ||\delta||_\infty \le \epsilon, \tag{5}$$

where $\mathcal{C}$ is the set of indices for all compromised clients. Since the resulting $\delta$ increases the probability that the model classifies any data as the target label, it can be viewed as the features of the target label learned by the model. Moreover, the constraint $||\delta||_\infty \le \epsilon$ is employed to regulate the influence of $\delta$ on the semantic content of $x$. Without this constraint, the solution to Equation 5 could become overly conspicuous, potentially obscuring the original semantic information of $x$. However, a fixed $\delta$ may limit attack effectiveness and make *Bad-PFL* more detectable. Therefore, instead of seeking a static $\delta$, we desire to create a dynamic trigger that can adapt to different input samples. This adaptability not only enhances *Bad-PFL*'s stealthiness but also enables the trigger to exploit specific characteristics of each sample, thereby improving attack performance. In practice, we employ a generative network parameterized by $w$, denoted as $\mathcal{G}_w$, which takes $x$ as input and produces a noise of the same shape as $x$. To control the output intensity of $\mathcal{G}_w$, we use $\tanh(\cdot)$ as the activation function of the final layer, scaling the output of $\mathcal{G}_w$ by multiplying it with $\epsilon$ to satisfy the constraint in Equation 5. Formally, we have $\delta = \epsilon \cdot \mathcal{G}_w(x)$.

**Craft disruptive noise.** In the second step, *Bad-PFL* generates a sample-specific disruptive noise $\xi$ for $x$, which is achieved by solving the following optimization problem:

$$\xi = \arg\max_{\xi} \left[\mathcal{L}(F(x + \xi; \theta_g), y)\right], s.t., ||\xi||_\infty \le \sigma. \tag{6}$$

Equation 6 seeks to identify a noise pattern that maximizes the loss function when added to $x$. To illustrate, think of it as introducing a layer of distortion to an image. Just as distortion can obscure the original image, the noise $\xi$ complicates the model's ability to accurately discern the ground-truth characteristics of $x$. We approximately solve Equation 6 by setting $\xi$ to $\sigma \cdot \text{sign}(\nabla_x \mathcal{L}(F(x; \theta_g), y))$, where $\text{sign}(\cdot)$ is an element-wise operation that returns the sign of the inputs.

**The ultimate local training process of compromised clients.** The parameters $w$ in the generative model need to be optimized to fit the global model. As shown in Algorithm 1, whenever a compromised client is selected, it first optimizes the generative network to produce $\delta$ that aligns with features of target category $y_t$ learned by the global model:

$$\begin{aligned} \min_{w} \quad & \mathbb{E}_{(x,y)\sim D_i, i\in\mathcal{C}} \left[\mathcal{L}(F(x + \mathcal{T}(x); \theta_g), y_t)\right] \\ = & \mathbb{E}_{(x,y)\sim D_i, i\in\mathcal{C}} \left[\mathcal{L}(F(x + \epsilon \cdot \mathcal{G}_w(x) + \sigma \cdot \text{sign}(\nabla_x \mathcal{L}(F(x; \theta_g), y)); \theta_g), y_t)\right]. \end{aligned} \tag{7}$$

The compromised client resolves Equation 4 to strengthen the global model's capacity to recognize and respond to the trigger. It subsequently trains its personalized model and finally uploads its local backdoored model to the server. For the sake of brevity, we place the explanation of how *Bad-PFL* addresses the three issues mentioned in Section 2.2 to Appendix D.

# 4 EMPIRICAL EVALUATION

## 4.1 SETUP

**Datasets and models.** We evaluate on three benchmark datasets: SVHN, CIFAR-10, and CIFAR-100, using ResNet10 as the default model. Appendix B.2 examines the effectiveness of *Bad-PFL* across varying model sizes (ResNet18, ResNet34) and architectures (MobileNet, DenseNet).

**FL settings.** Following existing studies (Zhuang et al., 2024), we set 100 clients and 1000 training rounds, with 10 clients being compromised. During each training round, 10% of clients are randomly selected. To simulate a non-IID setting, we use a Dirichlet distribution with a factor of 0.5

for data sampling. Each client trains their local and personalized models using SGD with a learning rate of 0.1 and a batch size of 32 for 15 steps (roughly one epoch).

**PFL methods and baseline attacks.** We adopt seven mainstream PFL methods to evaluate the performance of our attack, including FedProx (Li et al., 2020), SCAFFOLD (Karimireddy et al., 2020), Ditto (Li et al., 2021a), FedBN (Li et al., 2021b), FedRep (Collins et al., 2021), FedPAC (Xu et al., 2023). To facilitate a comparative study, we include six state-of-the-art backdoor attacks: DBA (Xie et al., 2020), FCBA (Liu et al., 2024), ModRep (Bagdasaryan et al., 2020), PGD-Bkd (Wang et al., 2020a), Neurotoxin (Zhang et al., 2022), and LF-Attack (Zhuang et al., 2024).

**Defenses.** We apply various backdoor defenses, including ClipAvg (Wang et al., 2020b), Multi-Krum (Blanchard et al., 2017), Median (Fang et al., 2020), Sign (Guo et al., 2023), NAD (Li et al., 2021c), I-BAU (Zeng et al., 2022), and Fine-tuning (FT) (Chen et al., 2022).

**Metrics.** We report average accuracy (Acc, %) over clean samples and attack success rate (ASR, %) over triggered samples for clients' personalized models on their test sets.

**Others.** All attacks use a poisoning rate $\alpha$ of 0.2. See Appendix A for hyperparameters of PFL methods, baseline attacks, defenses, and specifics of our generative network. Appendix C discusses the attack cost of *Bad-PFL*. For *Bad-PFL*, we adopt $\epsilon = \sigma = \frac{4}{255}$. The compromised clients utilize the Adam optimizer with a learning rate of 0.01 to train generative network for 30 steps. The target label $y_t$ is randomly generated. Appendix A provides examples of the non-IID partitioning, while Appendix B.2 shows the convergence curves of *Bad-PFL*, the impact of $\epsilon$ and $\sigma$ on attack performance, visualizations of the triggers, and a discussion on the attack costs. *The source codes can be found in supplementary material and will be released upon the acceptance of this paper.*

## 4.2 THE ATTACK PERFORMANCE IN PFL

Table 1: Attack performance comparison of various schemes over CIFAR10. We bold the best result and underline the runner-up.

| Attack | FedAvg Acc | FedAvg ASR | SCAFFOLD Acc | SCAFFOLD ASR | FedProx Acc | FedProx ASR | Ditto Acc | Ditto ASR | FedBN Acc | FedBN ASR | FedRep Acc | FedRep ASR | FedPAC Acc | FedPAC ASR |
|---|---|---|---|---|---|---|---|---|---|---|---|---|---|---|
| ModRep | 68.00 | 63.81 | 79.04 | 54.41 | 76.57 | 37.58 | 77.42 | 70.04 | 81.91 | 27.52 | 79.99 | 23.38 | 82.49 | 38.21 |
| Neurotoxin | 79.75 | 80.53 | 80.09 | 79.48 | 76.85 | 71.30 | 78.76 | 69.00 | 81.11 | 59.48 | 79.83 | 28.41 | 81.33 | 82.10 |
| PGD-Bkd | 79.27 | 78.61 | 78.95 | 94.19 | 77.04 | 74.54 | 78.72 | 72.23 | 81.21 | 54.81 | 79.77 | 20.25 | 81.54 | 63.45 |
| DBA | 78.97 | 92.41 | 79.11 | 94.85 | 77.47 | 83.59 | 79.70 | 76.45 | 80.36 | 31.33 | 79.80 | 15.54 | 82.46 | 18.23 |
| FCBA | 78.92 | 97.33 | 77.82 | **99.94** | 77.24 | 88.89 | 78.11 | 79.20 | 80.54 | 37.88 | 81.00 | 16.91 | 83.06 | 19.91 |
| LF-Attack | 79.85 | 95.90 | 78.86 | 95.98 | 76.82 | 85.46 | 78.20 | 78.94 | 81.09 | 44.55 | 80.90 | 12.82 | 83.25 | 15.82 |
| *Bad-PFL* | 79.28 | **99.88** | 78.95 | 99.80 | 77.36 | **99.68** | 78.88 | **94.12** | 80.72 | **82.22** | 80.29 | **97.95** | 82.67 | **99.10** |

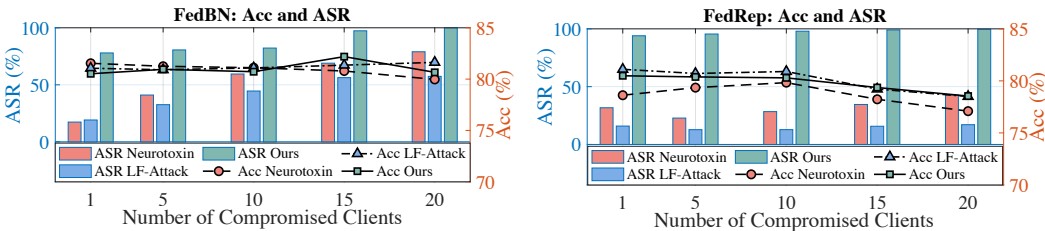

Figure 2: Preformance comparison of varying compromised client numbers for FedBN and FedRep.

Table 1 reports the performance of seven backdoor attack methods across seven PFL methods on CIFAR-10. Due to space constraints and the similarity of results obtained from SVHN and CIFAR-100, we leave those experimental results in Appendix B.2. As shown in Table 1, all attack-PFL combinations achieve comparable accuracy across personalized models. This indicates that the attack methods employed are stealthy, leading us to focus primarily on analyzing ASRs. Notably, we observe significant fluctuations in the baseline attacks when applied to different PFL methods, particularly those partial model-sharing methods, including FedBN, FedRep, and FedPAC. In these cases, the baseline attacks are often hard to inject backdoors into personalized models, as evidenced by low ASRs. In contrast, *Bad-PFL* demonstrates much more consistent and superior performance, achieving an impressive ASR of at least 80% across all cases.

Table 2: Performance comparison of various backdoor attacks against robust aggregation defenses in CIFAR-10. We bold the best result for FedBN and FedRep settings, respectively.

| Attack | PFL | ClipAvg | | Multi-Krum | | Median | | Sign | |
|--------|-----|------|------|------|------|------|------|------|------|
| | | Acc | ASR | Acc | ASR | Acc | ASR | Acc | ASR |
| ModRep | | 75.07 | 24.21 | 66.66 | 17.43 | 75.56 | 19.08 | 31.91 | 14.72 |
| Neurotoxin | | 83.19 | 54.93 | 63.59 | 38.97 | 71.76 | 53.82 | 31.93 | 20.03 |
| PGD-Bkd | | 82.76 | **94.34** | 65.61 | 73.47 | 73.68 | **64.01** | 31.33 | 14.94 |
| DBA | FedBN | 83.66 | 30.90 | 67.93 | 28.42 | 72.68 | 30.55 | 32.04 | 12.22 |
| FCBA | | 82.64 | 37.52 | 67.06 | 34.54 | 74.79 | 34.95 | 31.74 | 15.01 |
| LF-Attack | | 82.62 | 43.60 | 66.38 | 22.90 | 73.38 | 29.10 | 30.92 | 17.10 |
| *Bad-PFL* | | 82.55 | 82.66 | 66.93 | **80.28** | 74.44 | 50.52 | 31.42 | **24.13** |
| ModRep | | 77.17 | 20.59 | 68.95 | 17.32 | 69.99 | 12.14 | 33.29 | 15.18 |
| Neurotoxin | | 78.42 | 20.40 | 66.14 | 27.18 | 72.14 | 23.41 | 35.60 | 19.15 |
| PGD-Bkd | | 80.32 | 17.59 | 72.45 | 18.45 | 68.58 | 16.64 | 36.79 | 10.90 |
| DBA | FedRep | 78.98 | 14.80 | 71.19 | 15.26 | 69.66 | 12.97 | 37.72 | 12.32 |
| FCBA | | 82.72 | 16.08 | 71.37 | 16.17 | 71.22 | 13.08 | 32.89 | 13.17 |
| LF-Attack | | 81.99 | 11.96 | 70.93 | 12.65 | 70.96 | 12.06 | 34.03 | 12.54 |
| *Bad-PFL* | | 81.59 | **97.28** | 70.41 | **96.15** | 70.23 | **77.21** | 34.49 | **20.32** |

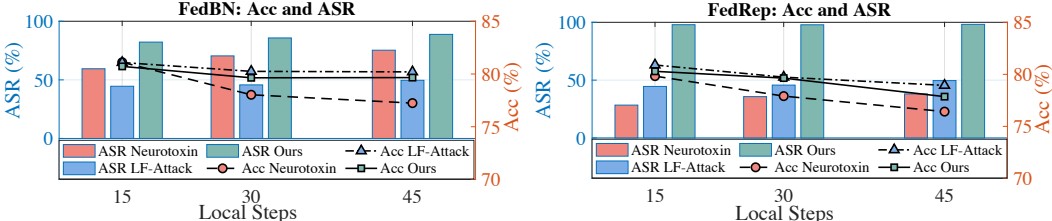

Figure 3: Preformance comparison from different local steps for both FedBN and FedRep.

Table 3: Backdoor persistence comparison of various attacks over CIFAR10. We bold the best result. FT-15, FT-30, FT-45 denote Fine-tuning with 15, 30, and 45 steps respectively. We bold the best result for FedBN and FedRep settings, respectively.

| Attack | PFL | Before | | NAD | | I-BAU | | FT-15 | | FT-30 | | FT-45 | |
|--------|-----|------|------|------|------|------|------|------|------|------|------|------|------|
| | | Acc | ASR | Acc | ASR | Acc | ASR | Acc | ASR | Acc | ASR | Acc | ASR |
| Neurotoxin | | 81.11 | 59.48 | 75.97 | 52.35 | 79.26 | 50.54 | 82.59 | 40.02 | 82.79 | 32.07 | 83.06 | 18.35 |
| LF-Attack | FedBN | 81.09 | 44.55 | 76.69 | 39.83 | 79.75 | 25.76 | 81.86 | 32.30 | 82.01 | 25.45 | 82.27 | 18.69 |
| *Bad-PFL* | | 80.72 | **82.22** | 76.75 | **76.24** | 77.78 | **75.15** | 81.79 | **81.12** | 82.04 | **80.49** | 82.21 | **80.12** |
| Neurotoxin | | 79.83 | 28.41 | 74.95 | 25.86 | 78.71 | 12.36 | 80.75 | 22.03 | 81.09 | 17.15 | 81.27 | 16.24 |
| LF-Attack | FedRep | 80.90 | 12.82 | 75.74 | 11.91 | 78.65 | 11.52 | 81.33 | 12.58 | 81.82 | 12.35 | 81.88 | 11.82 |
| *Bad-PFL* | | 80.29 | **97.95** | 76.33 | **86.25** | 79.58 | **76.58** | 80.67 | **97.31** | 80.79 | **97.76** | 81.10 | **97.01** |

## 4.3 THE ATTACK PERFORMANCE AGAINST STATE-OF-THE-ART DEFENSES

Section 4.2 primarily focuses on the FedAvg aggregator, where each client's contribution is treated equally during aggregation. Several robust aggregation methods have been proposed to identify and downweight or eliminate the influence of compromised clients in aggregation. However, it is stressed that the local models of different clients are inherently different in the non-IID setting, which can lead to incorrect identification and degrade Acc. Here, we specifically examine FedBN and FedRep, as they demonstrate the best defensive performance in Section 4.2. Table 2 presents the attack results against robust aggregation methods. As can be seen, ClipAvg maintains accuracy better than others, but it can only defend against simpler backdoor attack methods, such as ModRep. Its defense performance is limited against more sophisticated attacks like Neurotoxin and PGD-Bkd. Furthermore, Sign can effectively counter almost all attack methods; however, the gradient quantization makes the model challenging to train. Among the four defense methods, Median performs the best, successfully mitigating most backdoor attacks while maintaining accuracy in most cases. Nonetheless, even with Median, the ASRs of *Bad-PFL* remain high at 50.52% and 77.21%, highlighting the significant attack effectiveness of *Bad-PFL*.

## 4.4 BACKDOOR PERSISTENCE

The persistence of embedded backdoors can be examined from two angles: first, by reducing the number of compromised clients in FL, which extends the expected time for a compromised client to

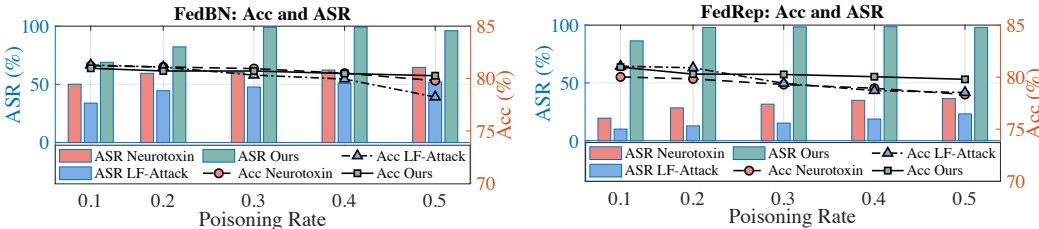

Figure 4: Preformance comparison under varying poisoning rates for FedBN and FedRep.

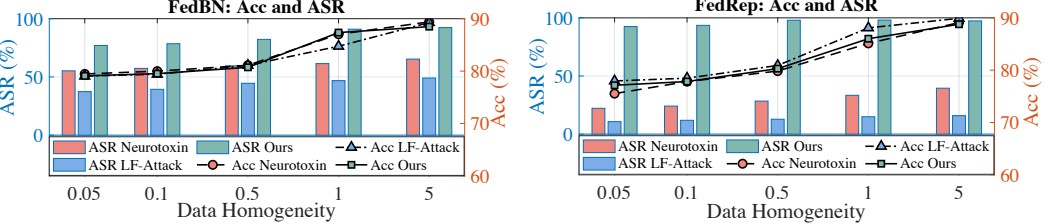

Figure 5: Attack performance comparison with varying data heterogeneity degree.

be selected; and second, by allowing clients to use fine-tuning or backdoor removal methods after the FL process is completed. Figure 2 and Table 3 present the corresponding results.

Overall, as the number of compromised clients increases, it becomes easier to insert backdoors into personalized models. Notably, when there is only one compromised client, the server is expected to select the compromised client once every ten rounds. At this point, we observe that both Neurotoxin and LF-Attack yield low ASRs while *Bad-PFL* can achieve a remarkable 94.01% ASR. Additionally, we find that as the number of compromised clients increases, the Acc tends to decline. This is reasonable, as a higher number of compromised clients indicates a lower proportion of clean data.

We include three client-side backdoor defenses: NAD, I-BAU, and FT. As can be observed in Table 3, these defense methods provide some mitigation; however, they struggle to effectively erase backdoors, especially against *Bad-PFL*, which maintains ASRs of over 70%. In fact, all three defenses involve fine-tuning the model on natural data. However, the backdoor embedded by *Bad-PFL* arises naturally from natural data, making the backdoor inherently durable when trained in natural data and difficult to remove using these techniques.

### 4.5 SENSITIVITY ANALYSIS AND ABLATION STUDY

We here examine the contribution of $\delta$ and $\xi$ to attack performance, the local steps of clients, the degree of data heterogeneity among clients, and the poisoning rate of compromised clients.

**The contribution of $\delta$ and $\xi$.** Table 4 shows the impact of different components of our trigger on attack performance. It is observed that removing the target feature $\delta$ leads to a significant drop in attack performance, as $x + \xi$ no longer contains features about the target class. When we remove $\xi$, there is a decline in attack performance, which can be attributed to the fact

Table 4: Ablation study of $\delta$ and $\xi$.

| Component | FedBN | | FedRep | |
|---|---|---|---|---|
| | Acc | ASR | Acc | ASR |
| w/o $\delta$ | 80.78 | 13.74 | 80.17 | 12.82 |
| w/o $\xi$ | 80.56 | 68.56 | 80.20 | 79.32 |
| Both | 80.72 | **82.22** | 80.29 | **97.95** |

that $x + \delta$ still retains features of the ground-truth label, allowing it to potentially be classified into the corresponding ground-truth label category. Refer to Appendix E for a more detailed exploration of the relationship between $\delta$ and $\xi$.

**The impact of local steps.** Figure 3 illustrates the performance of various attack methods across different local steps of clients. Apart from *Bad-PFL*, which consistently outperforms the two baseline methods, the overall trend is that as local steps increase, ASR gradually rise while Acc declines. As the number of local steps increases, the client models begin to overfit their local datasets, leading to a decrease in generalization performance. The slight increase in attack performance is likely due to the backdoor becoming more deeply ingrained in the local model after more training steps. Consequently, both the global model and personalized models become further compromised.

**The impact of poisoning rate.** Figure 4 shows the performance of different attack methods at varying poisoning rates for FedBN and FedRep. A higher poisoning rate tends to optimize the backdoor task at the expense of the main task, ultimately resulting in a higher ASR and lower Acc. The results in Figure 4 validate this.

**The degree of data heterogeneity.** The degree of data heterogeneity among clients can be controlled by the parameters of Dirichlet distribution, where lower values represent higher heterogeneity. Figure 5 presents the attack performance of various methods under different levels of data heterogeneity. Appendix A provides examples of the distribution of client heterogeneity at different parameters. Generally, as heterogeneity decreases, the solutions of different client models tend to be the same. As expected, in Figure 5, with increasing IID levels, both accuracy and ASR improve.

## 5 RELATED WORK

**Personalized federated learning.** PFL methods can be divided into full model-sharing and partial model-sharing methods. FedProx (Li et al., 2020) introduces a proximal term to the local training objective to manage the distance between the local and the global model. SCAFFOLD (Karimireddy et al., 2020) uses control variables to calibrate the updates of the local model. Fine-tuning (Chen et al., 2022) is sometimes also considered a technique for full model sharing. Ditto (Li et al., 2021a) adds a regularization term that encourages the personalized model to remain close to the global model. Partial model sharing involves decoupling the personalized model parameters into shared and private ones, with the shared ones submitted to the server for aggregation. FedBN (Li et al., 2021b) privatizes the batch normalization layers, while the remaining layers are updated according to FedAvg protocol. FedRep (Collins et al., 2021) shares the feature extraction layers while keeping the classification head private to fit local datasets. FedPAC (Xu et al., 2023) suggests weighting the private classification heads from various clients to form the final classifier. In addition, Scott et al. (2024) proposed training a hypernet that can customize a model based on the characteristics of each client. Moreover, some studies (Scott et al., 2024) proposed hypernet-based methods, where a hypernet is trained to produce a model based on each client's characteristics.

**Federated backdoor attack and defenses.** Federated backdoor attacks are often executed by having compromised clients upload local backdoored models, with variations arising in how these models are constructed. ModRep (Bagdasaryan et al., 2020) amplifies the parameters of the backdoored model, enabling it to dominate the aggregation process. DBA (Xie et al., 2020) splits a global trigger into multiple sub-triggers, with each compromised client holding one of these sub-triggers. PGD-Bkd (Wang et al., 2020a) restricts the parameter changes of the backdoored model to evade robust aggregation methods and detection methods. Neurotoxin (Zhang et al., 2022) embeds backdoors within parameters that have smaller update magnitudes, increasing the persistence of the backdoor. LF-Attack (Zhuang et al., 2024) focuses on backdoor-critical layers to enhance attack effectiveness. Iba (Nguyen et al., 2024) uses generators to produce triggers, while Perdoor (Alam et al., 2022) crafts triggers tailored to specific neurons. Although some studies (Lai et al., 2022; Lyu et al., 2024) explored backdoor attacks against PFL methods, these tend to focus too narrowly on specific PFL methods or are easily countered by common defense mechanisms. For instance, PFedBA (Lyu et al., 2024) employs fixed trigger patterns, making them detectable. To counter such attacks, robust aggregation methods can be utilized, and clients can also perform backdoor elimination after FL concludes. Common methods for the former include median (Fang et al., 2020), Krum (Blanchard et al., 2017), etc. For elimination, most studies (Li et al., 2021c; Zeng et al., 2022; Chen et al., 2022) suggested diminishing the influence of less significant parameters on the model.

## 6 CONCLUSION

In this study, we identified three key factors that contribute to the failure of existing backdoor attacks in PFL methods. We developed *Bad-PFL* to address these issues by employing natural features as our trigger instead of manually designed ones. Models trained on natural data inherently learn the relationship between our trigger and the target label, implicitly embedding the backdoor within them. Our extensive experiments demonstrated the superior performance of *Bad-PFL* against mainstream attacks for PFL methods. We hope this study can correct the misconception among PFL community that existing PFL methods are immune to backdoor attacks.

## 7  ETHICS STATEMENT

This paper presents an attack method that undermines the trustworthiness of federated learning. Although this attack method may seem harmful, we strongly believe that the benefits of publishing this paper outweigh the drawbacks. Specifically, this attack method can motivate researchers to explore more effective defense strategies, serve as an assessment tool for testing the trustworthiness of federated learning, and raise awareness of potential threats faced by users implementing federated learning in real-world scenarios.

## 8  REPRODUCIBILITY

The source codes are available in https://github.com/fmy266/Bad-PFL.

ACKNOWLEDGMENTS

This work was supported by the National Natural Science Foundation of China under grant number 62202170.

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

# A THE DETAILS HYPERPARAMETERS ABOUT EXPERIMENT

**The detailed hyperparameters of PFL methods and baseline attacks.** For the PFL methods, we use the same training configuration as that of the local models to train personalized models. Please refer to Section 4.1 for the training settings of the local models. For Ditto and FedProx, we set the regularization strength $\mathcal{R}$ to 0.1. DBA allows each compromised client to select a specific size of trigger and arranges these triggers systematically on the images. We set the size of the triggers used by the compromised clients to $1 \times 3$, starting from the top left corner, with a gap of one pixel between each trigger. After filling five triggers per row, a new row is started, maintaining the same one-pixel gap. FCBA is an improved version of DBA, and we set its hyperparameter $m$ to 4. ModRep first amplifies the differences between the parameters of the compromised clients' local backdoored models and the global model before uploading them. In this paper, this difference is amplified by a factor of 10. PGD-Attack constrains the norm of the parameter differences between the local and global models within a specified value, which we set to 1. Neurotoxin primarily updates the parameters with the smallest update magnitudes. We choose to update the bottom 10% of parameters based on their update magnitudes. For LF-Attack, we set $\tau$ to 0.95 to identify the backdoor-critical layers, and we set $\lambda$ to 1.

**The detailed hyperparameters of defenses.** We detail the hyperparameters of defenses as follows:

Table 5: The architecture of our generative model.

| | | |
|---|---|---|
| Generator | Encoder | Conv2d + BatchNorm2d + Relu |
| | | Conv2d + BatchNorm2d + Relu |
| | | Conv2d + BatchNorm2d + Relu |
| | | Conv2d + BatchNorm2d + Relu |
| | Decoder | ConvTranspose2d + BatchNorm2d + Relu |
| | | ConvTranspose2d + BatchNorm2d + Relu |
| | | ConvTranspose2d + BatchNorm2d + Relu |
| | | ConvTranspose2d + BatchNorm2d + Tanh |

- ClipAvg builds upon FedAvg by incorporating a pruning operation on the model parameters uploaded by clients, ensuring that the strength of each parameter does not exceed a threshold t. We set $t$ to 1.

- The server calculates Krum distance known using models from the closest $N \times C - f$ clients, where $f$ is a hyperparameter assumed to be equal to or greater than the number of compromised clients. In our setting, the expected number of compromised clients selected by the server in each training round is 1, so we set $f$ to 1. Subsequently, MultiKrum selects five clients with the highest distance scores to aggregate the global model.

- Median aggregates the model parameters uploaded by clients by taking the median value.

- Sign first aggregates the models from the clients and then uses the sign of aggregated values to update the global model. We multiply the sign of aggregated values by 0.01 to update the global model.

**The specific architecture of our generative model.** Our generative model follows an encoder-decoder architecture, as shown in Table 5. The encoder comprises four convolutional layers, each followed by batch normalization ($\mathrm{BatchNorm2d}$) and $\mathrm{Relu}$ activation functions. Each convolutional layer doubles the depth, uses a kernel size of 4, a stride of 2, and padding of 1. The decoder replicates the encoder's structure using transpose convolutional layers, with the final layer employing a $\mathrm{tanh}$ activation function.

**Non-IID partition examples.** To provide a clearer understanding of the degree of non-IID distribution, we present the label distribution of the local training datasets for clients using different parameters of the Dirichlet distribution. Due to page limitations, we only include the label distribution for the local training datasets of the first 10 clients. We also report the empirical standard deviation of the label distribution in the clients' local training datasets. Standard deviation indicates how much the numbers in a set are spread out and thus can serve as a measure of data heterogeneity, facilitating a more intuitive understanding. A larger standard deviation indicates a higher level of data heterogeneity among the clients. Specific results can be found in Tables 6 to 10. The average standard deviation for the clients is 103.21, 90.48, 60.88, 45.15, and 25.24, corresponding to Dirichlet distribution parameters of 0.05, 0.1, 0.5, 1, and 5, respectively. When using a Dirichlet distribution with a parameter of 0.5, many clients may have only 1 to 2 samples for certain labels. In contrast, when using a Dirichlet distribution with a parameter of 5, there is a small variation in the number of samples for each label among clients. When employing a Dirichlet distribution with a parameter of 0.05, the data for clients is primarily concentrated in one or two categories.

## B  SUPPLEMENTARY EXPERIMENT

### B.1  EXPERIMENT ABOUT SECTION 2

We here provide experimental results mentioned in Section 2.2. Unless otherwise specified, the experimental configurations follow those outlined in Section 4.1 and Appendix A.

***In full model-sharing methods, the regularization term alone is inadequate for directly transferring the backdoor from the global model to personalized models.*** Table 11 reports attack results. Here, $G - Model$ and $P - Model$ refer to the global model and personalized models, respectively. $\mathcal{R}$ indicates the regularization term in Equation 2. GAvg. and PAvg. represent the averages of the data corresponding to the G-Model and P-Model, respectively. As we analyzed in Section 2.2,

Table 6: The number of samples for different labels in the local datasets of the first ten clients, generated using a Dirichlet distribution with a parameter of 0.05.

| Label | Airplane | Automobile | Bird | Cat | Deer | Dog | Frog | Horse | Ship | Truck | Std. |
|---|---|---|---|---|---|---|---|---|---|---|---|
| Client 0 | 8 | 198 | 29 | 9 | 38 | 139 | 16 | 5 | 4 | 54 | 65.97 |
| Client 1 | 313 | 0 | 0 | 6 | 6 | 8 | 8 | 0 | 157 | 2 | 104.26 |
| Client 2 | 4 | 0 | 0 | 0 | 12 | 24 | 459 | 0 | 0 | 1 | 143.92 |
| Client 3 | 13 | 0 | 0 | 82 | 20 | 55 | 13 | 309 | 0 | 8 | 94.89 |
| Client 4 | 166 | 0 | 0 | 0 | 305 | 29 | 0 | 0 | 0 | 0 | 103.51 |
| Client 5 | 0 | 0 | 10 | 2 | 3 | 71 | 414 | 0 | 0 | 0 | 129.76 |
| Client 6 | 27 | 316 | 0 | 16 | 53 | 32 | 36 | 4 | 0 | 16 | 95.01 |
| Client 7 | 3 | 0 | 0 | 0 | 13 | 20 | 412 | 0 | 46 | 6 | 128.01 |
| Client 8 | 19 | 5 | 74 | 7 | 31 | 34 | 10 | 301 | 2 | 17 | 90.69 |
| Client 9 | 15 | 9 | 254 | 20 | 28 | 97 | 24 | 8 | 18 | 27 | 76.09 |

Table 7: The number of samples for different labels in the local datasets of the first ten clients, generated using a Dirichlet distribution with a parameter of 0.1.

| Label | Airplane | Automobile | Bird | Cat | Deer | Dog | Frog | Horse | Ship | Truck | Std. |
|---|---|---|---|---|---|---|---|---|---|---|---|
| Client 0 | 24 | 6 | 217 | 25 | 48 | 44 | 62 | 3 | 71 | 0 | 63.65 |
| Client 1 | 0 | 46 | 0 | 59 | 15 | 44 | 0 | 333 | 3 | 0 | 102.01 |
| Client 2 | 1 | 174 | 0 | 208 | 3 | 22 | 51 | 40 | 1 | 0 | 76.91 |
| Client 3 | 1 | 97 | 24 | 15 | 22 | 0 | 117 | 1 | 223 | 0 | 73.87 |
| Client 4 | 4 | 18 | 14 | 14 | 49 | 26 | 132 | 25 | 27 | 191 | 61.53 |
| Client 5 | 0 | 0 | 10 | 0 | 442 | 43 | 2 | 0 | 3 | 0 | 138.37 |
| Client 6 | 9 | 35 | 18 | 19 | 38 | 12 | 14 | 20 | 18 | 317 | 94.27 |
| Client 7 | 0 | 8 | 0 | 11 | 6 | 388 | 77 | 5 | 5 | 0 | 120.99 |
| Client 8 | 273 | 68 | 0 | 36 | 2 | 0 | 0 | 109 | 12 | 0 | 86.59 |
| Client 9 | 2 | 132 | 266 | 60 | 7 | 1 | 1 | 7 | 24 | 0 | 86.58 |

Table 8: The number of samples for different labels in the local datasets of the first ten clients, generated using a Dirichlet distribution with a parameter of 0.5.

| Label | Airplane | Automobile | Bird | Cat | Deer | Dog | Frog | Horse | Ship | Truck | Std. |
|---|---|---|---|---|---|---|---|---|---|---|---|
| Client 0 | 39 | 8 | 1 | 5 | 71 | 42 | 53 | 26 | 204 | 51 | 58.80 |
| Client 1 | 20 | 11 | 3 | 42 | 98 | 5 | 258 | 8 | 22 | 33 | 78.26 |
| Client 2 | 26 | 12 | 37 | 79 | 44 | 127 | 2 | 10 | 116 | 47 | 43.80 |
| Client 3 | 41 | 292 | 22 | 14 | 10 | 50 | 2 | 11 | 17 | 41 | 86.49 |
| Client 4 | 10 | 120 | 6 | 24 | 6 | 56 | 187 | 54 | 4 | 33 | 59.88 |
| Client 5 | 37 | 64 | 25 | 216 | 8 | 88 | 3 | 22 | 36 | 1 | 64.42 |
| Client 6 | 43 | 69 | 17 | 0 | 4 | 110 | 1 | 65 | 84 | 107 | 43.14 |
| Client 7 | 37 | 4 | 186 | 225 | 6 | 33 | 3 | 1 | 1 | 4 | 83.51 |
| Client 8 | 130 | 7 | 8 | 49 | 83 | 90 | 84 | 34 | 9 | 6 | 44.39 |
| Client 9 | 2 | 1 | 86 | 85 | 19 | 130 | 25 | 34 | 101 | 17 | 46.14 |

Table 9: The number of samples for different labels in the local datasets of the first ten clients, generated using a Dirichlet distribution with a parameter of 1.

| Label | Airplane | Automobile | Bird | Cat | Deer | Dog | Frog | Horse | Ship | Truck | Std. |
|---|---|---|---|---|---|---|---|---|---|---|---|
| Client 0 | 45 | 6 | 5 | 21 | 8 | 35 | 21 | 56 | 133 | 170 | 56.75 |
| Client 1 | 54 | 3 | 18 | 82 | 33 | 32 | 32 | 27 | 138 | 81 | 40.06 |
| Client 2 | 5 | 21 | 22 | 95 | 22 | 8 | 53 | 162 | 3 | 109 | 54.23 |
| Client 3 | 60 | 185 | 51 | 38 | 2 | 72 | 72 | 11 | 5 | 4 | 55.18 |
| Client 4 | 51 | 17 | 80 | 9 | 60 | 6 | 2 | 221 | 26 | 28 | 65.24 |
| Client 5 | 67 | 62 | 38 | 113 | 98 | 3 | 2 | 19 | 79 | 19 | 39.62 |
| Client 6 | 36 | 82 | 33 | 63 | 17 | 28 | 60 | 56 | 69 | 56 | 20.50 |
| Client 7 | 1 | 24 | 92 | 113 | 27 | 18 | 105 | 78 | 2 | 40 | 42.92 |
| Client 8 | 131 | 40 | 12 | 21 | 90 | 54 | 47 | 58 | 2 | 45 | 37.95 |
| Client 9 | 84 | 38 | 66 | 21 | 106 | 40 | 10 | 110 | 18 | 7 | 39.02 |

the regularization term alone is insufficient for transferring the backdoor from the global model to personalized models unless the backdoor-critical parameters are weighted to compel personalized models to learn the backdoor.

Table 10: The number of samples for different labels in the local datasets of the first ten clients, generated using a Dirichlet distribution with a parameter of 5.

| Label | Airplane | Automobile | Bird | Cat | Deer | Dog | Frog | Horse | Ship | Truck | Std. |
|---|---|---|---|---|---|---|---|---|---|---|---|
| Client 0 | 27 | 75 | 43 | 63 | 34 | 31 | 104 | 40 | 54 | 29 | 24.64 |
| Client 1 | 137 | 31 | 66 | 50 | 28 | 33 | 36 | 66 | 13 | 40 | 34.77 |
| Client 2 | 47 | 74 | 55 | 59 | 64 | 48 | 41 | 28 | 60 | 24 | 15.75 |
| Client 3 | 26 | 51 | 27 | 21 | 38 | 78 | 37 | 111 | 38 | 73 | 28.75 |
| Client 4 | 50 | 31 | 117 | 29 | 50 | 65 | 63 | 20 | 22 | 53 | 28.63 |
| Client 5 | 45 | 52 | 47 | 15 | 45 | 35 | 113 | 36 | 39 | 73 | 26.52 |
| Client 6 | 31 | 61 | 103 | 21 | 46 | 49 | 43 | 22 | 61 | 63 | 24.20 |
| Client 7 | 22 | 55 | 41 | 36 | 50 | 68 | 19 | 94 | 57 | 58 | 22.16 |
| Client 8 | 53 | 46 | 81 | 9 | 88 | 24 | 51 | 56 | 44 | 48 | 23.25 |
| Client 9 | 73 | 35 | 87 | 66 | 26 | 29 | 68 | 27 | 25 | 64 | 23.73 |

Table 11: ASR(%) comparison between G-Model and P-Model under non-IID setting. G-model and P-Model denote global model and personalized model, respectively.

| Attack | Strategy | FedProx | | Ditto | | pFedMe | | GAvg. | PAvg. |
|---|---|---|---|---|---|---|---|---|---|
| | | G-Model | P-Model | G-Model | P-Model | G-Model | P-Model | | |
| Neurotoxin | Vanilla | 94.08 | 71.30 | 95.02 | 69.00 | 90.26 | 77.90 | 93.12 | 72.73 |
| | w.o $\mathcal{R}$ | 92.38 | 13.99 | 95.49 | 10.20 | 90.12 | 10.86 | 92.66 | 11.68 |
| | Weighted $\mathcal{R}$ | 92.11 | 90.50 | 95.70 | 93.20 | 91.33 | 89.07 | 93.05 | 90.92 |
| PGD-Backdoor | Vanilla | 93.36 | 74.54 | 92.46 | 72.23 | 93.24 | 80.23 | 93.02 | 75.67 |
| | w.o $\mathcal{R}$ | 90.99 | 10.45 | 92.10 | 10.63 | 92.45 | 11.63 | 91.85 | 10.90 |
| | Weighted $\mathcal{R}$ | 92.59 | 88.93 | 92.19 | 90.82 | 92.60 | 90.29 | 92.46 | 90.01 |

Table 12: ASR (%) comparison across backdoor attack configurations.

| Attack | Strategy | FedBN | FedRep | Avg. |
|---|---|---|---|---|
| Neurotoxin | First Configuration | 59.48 | 28.41 | 43.95 |
| | Second Configuration | 96.89 | 98.72 | 97.81 |
| | Third Configuration | 10.66 | 10.52 | 10.59 |
| PGD-Backdoor | First Configuration | 54.81 | 20.25 | 37.53 |
| | Second Configuration | 97.96 | 96.01 | 96.99 |
| | Third Configuration | 10.06 | 11.07 | 10.57 |

Table 13: ASR (%) comparison over various models.

| PFL Method | FedBN | | FedRep | |
|---|---|---|---|---|
| Number of comprimised clients | Neurotoxin | PGD-Backdoor | Neurotoxin | PGD-Backdoor |
| 1 | 68.41 | 84.81 | 75.31 | 76.86 |
| 3 | 77.24 | 90.64 | 77.81 | 79.75 |
| 5 | 84.63 | 92.51 | 88.70 | 88.40 |
| 7 | 87.99 | 92.56 | 91.42 | 90.85 |
| 10 | 92.14 | 93.95 | 95.29 | 95.20 |
| 15 | 95.99 | 94.75 | 98.31 | 98.41 |

***In partial model-sharing methods, non-shared parameters hinder the backdoor effect, making it challenging for the backdoor to influence the decision-making processes of personalized models.*** Table 12 presents the attack performance of two backdoor attack methods under three different configurations for two partial model-sharing methods. The first configuration simply applies the backdoor attack methods to FedBN and FedRep. The second configuration fixes the shared parameters and fine-tunes only the non-shared parameters on the trigger-embedded and clean data for one epoch (15 steps). The third configuration is similar to the second but trains the model solely on clean data. The experimental analysis has already been discussed in Section 2.2, so we will not repeat it here.

***During the training process, the backdoor may be progressively diluted.*** In Section 2.2, we primarily discuss two scenarios in which the backdoor can be gradually removed. Here, we validate the first scenario, which posits that if compromised clients are selected by the server with a significant time gap, the backdoor may gradually be erased due to updates from benign models. Table 13 reports the effectiveness of various attack methods against FedBN and FedRep under the IID setting with varying numbers of compromised clients. Figure 2 (Section 4.4) illustrates the effectiveness

of different attack methods against FedBN and FedRep under the non-IID condition with varying numbers of compromised clients. For comparison, we observe that under the IID setting, the ASRs on FedBN-Neurotoxin pair are higher and more stable; for example, the ASR only decreases by about 20% when reducing the number of compromised clients from 10 to 1. In contrast, under the Nnon-IID setting, the ASR drops by nearly 50%. This indicates that the backdoor is more easily diluted under the non-IID condition. The second conclusion can be validated by the results (Table 3) in Section 4.4. When clients fine-tune the model on their own datasets (FT-15, FT-30, FT-45), the ASRs significantly decrease.

## B.2 EXPERIMENT ABOUT SECTION 4

In this section, unless explicitly stated otherwise, we follow the experimental configuration used in Section 4.2.

Table 14: Attack performance comparison over various models. We bold the best result.

| Attack | PFL | ResNet10 | | ResNet18 | | ResNet34 | | MobileNetV2 | | DenseNet | |
|---|---|---|---|---|---|---|---|---|---|---|---|
| | | Acc | ASR | Acc | ASR | Acc | ASR | Acc | ASR | Acc | ASR |
| Neurotoxin | | 81.11 | 59.48 | 81.35 | 60.64 | 82.63 | 65.97 | 70.36 | 36.54 | 77.60 | 43.91 |
| LF-Attack | FedBN | 81.09 | 44.55 | 81.21 | 45.22 | 82.24 | 50.48 | 70.95 | 29.54 | 78.22 | 32.58 |
| *Bad-PFL* | | 80.72 | **82.22** | 80.98 | **86.85** | 82.07 | **90.12** | 70.10 | **75.17** | 77.39 | **79.98** |
| Neurotoxin | | 79.83 | 28.41 | 80.00 | 45.22 | 80.68 | 46.48 | 70.22 | 16.27 | 76.46 | 23.22 |
| LF-Attack | FedRep | 80.90 | 12.82 | 81.02 | 13.58 | 81.23 | 15.94 | 90.65 | 11.95 | 77.60 | 13.96 |
| *Bad-PFL* | | 80.29 | **97.95** | 80.52 | **98.54** | 80.76 | **99.30** | 70.03 | **84.21** | 76.34 | **94.22** |

### B.2.1 THE ATTACK PERFORMANCE OF *Bad-PFL* OVER DIFFERENT MODEL SIZES AND ARCHITECTURES.

Table 14 reports the effectiveness of different attack methods against FedBN and FedRep on ResNet10, ResNet18, ResNet34, MobileNetV2, and DenseNet. Overall, we observe that *Bad-PFL* achieves high effectiveness across all models while maintaining competitive Acc. Additionally, the order of model capacity from largest to smallest is ResNet34, ResNet18, ResNet10, DenseNet, and MobileNetV2. We find that larger models have more capacity for learning, resulting in higher Acc. However, increased model capacity also leaves more room for the implantation of backdoors, as indicated by the higher ASRs. This observation is consistent with conclusions drawn in existing literature.

Table 15: The performance of various backdoor attack methods in ViT. Here we employ FedRep.

| Attack | Neurotoxin | LF-Attack | *Bad-PFL* |
|---|---|---|---|
| Acc | 85.43 | 86.06 | 85.89 |
| ASR | 50.64 | 20.58 | 98.94 |

### B.2.2 ATTACK PERFORMANCE IN TRANSFORMERS

To further validate the versatility of *Bad-PFL* across different model architectures, we evaluate *Bad-PFL*'s performance against Vision Transformer (ViT). Specifically, we use a ViT pre-trained on ImageNet (provided by TorchVision) as the initialization for the server, with the classification head reinitialized to accommodate CIFAR-10. We employ FedRep for this evaluation. Table 15 reports the attack results of various methods on ViT. Due to its pre-training, ViT achieves approximately 5 points higher accuracy compared to convolutional neural network architectures. Additionally, our method performs well on ViT, achieving an ASR of 98.94%.

### B.2.3 THE EVALUATION RESULTS IN SVHN AND CIFAR-100

Tables 16 and 17 report the attack results of different methods against various PFL methods on CIFAR-100 and SVHN. In addition to the clear conclusion that *Bad-PFL* yields higher ASRs over baseline attacks, we have the following observation. We note that on SVHN, the different attack methods enjoy better attack results compared to CIFAR-10. We speculate that this is due to SVHN being a simpler dataset than CIFAR-10. Similarly, CIFAR-100 can be considered a fine-grained version of CIFAR-10, which is more challenging. On CIFAR-100, we observe lower Acc and ASR.

Table 16: Attack performance comparison of various schemes over CIFAR100. We bold the best result and underline the runner-up.

| Attack | SCAFFOLD | | FedProx | | Ditto | | FedBN | | FedRep | | FedPAC | |
|---|---|---|---|---|---|---|---|---|---|---|---|---|
| | Acc | ASR | Acc | ASR | Acc | ASR | Acc | ASR | Acc | ASR | Acc | ASR |
| Model-Replacement | 45.77 | 50.82 | 47.02 | 25.04 | 47.56 | 32.02 | 50.83 | 31.59 | 47.22 | 17.85 | 49.94 | 32.84 |
| Neurotoxin | 49.50 | 77.46 | 47.40 | 64.47 | 48.76 | 65.15 | 51.41 | 38.25 | 47.03 | 19.82 | 49.52 | 46.85 |
| PGD-Backdoor | 49.24 | 92.23 | 48.73 | 55.58 | 49.68 | 58.70 | 51.16 | 36.27 | 48.48 | 24.04 | 52.61 | 55.57 |
| DBA | 50.00 | 92.47 | 48.38 | 55.64 | 49.00 | 56.95 | 51.60 | 28.69 | 49.96 | 9.75 | 52.40 | 18.42 |
| FCBA | 49.62 | 99.98 | 49.83 | 59.99 | 50.53 | 60.30 | 51.06 | 45.73 | 49.33 | 11.39 | 52.33 | 19.99 |
| LF-Attack | 49.40 | 93.90 | 48.62 | 56.00 | 50.31 | 58.16 | 50.73 | 37.56 | 48.75 | 11.52 | 53.34 | 18.95 |
| *Bad-PFL* | 49.72 | 99.08 | 48.15 | **99.15** | 49.25 | **99.16** | 51.96 | **93.96** | 48.49 | **94.80** | 51.38 | **98.73** |

Table 17: Attack performance comparison of various schemes over SVHN. We bold the best result and underline the runner-up.

| Attack | SCAFFOLD | | FedProx | | Ditto | | FedBN | | FedRep | | FedPAC | |
|---|---|---|---|---|---|---|---|---|---|---|---|---|
| | Acc | ASR | Acc | ASR | Acc | ASR | Acc | ASR | Acc | ASR | Acc | ASR |
| Model-Replacement | 90.44 | 82.21 | 86.18 | 58.51 | 87.47 | 83.67 | 94.21 | 41.46 | 89.81 | 24.61 | 90.43 | 47.83 |
| Neurotoxin | 93.16 | 93.42 | 93.43 | 61.13 | 94.36 | 81.10 | 94.02 | 48.82 | 92.36 | 24.89 | 93.12 | 57.05 |
| PGD-Backdoor | 93.91 | 98.78 | 93.49 | 63.23 | 93.77 | 83.42 | 93.93 | 30.43 | 93.54 | 25.20 | 93.68 | 58.68 |
| DBA | 93.69 | 98.73 | 93.34 | 61.96 | 94.69 | 79.15 | 93.77 | 34.31 | 93.46 | 20.61 | 93.99 | 29.46 |
| FCBA | 93.77 | **99.54** | 93.50 | 64.00 | 93.87 | 82.30 | 94.08 | 45.06 | 93.49 | 21.06 | 94.12 | 30.00 |
| LF-Attack | 93.72 | 97.16 | 93.04 | 77.41 | 94.00 | 85.24 | 93.80 | 38.38 | 92.91 | 20.64 | 93.06 | 29.03 |
| *Bad-PFL* | 93.00 | 98.89 | 92.64 | **96.88** | 93.43 | **98.88** | 94.20 | **99.52** | 93.77 | **97.39** | 94.37 | **97.46** |

### B.2.4 THE IMPACT OF THE MAGNITUDE $\epsilon$ AND $\xi$ IN ATTACK PERFORMANCE OF *Bad-PFL*

Table 18 reports the ASRs of *Bad-PFL* when either $\delta$ or $\xi$ is fixed while varying the magnitude of the other. We observe that *Bad-PFL* is more sensitive to changes in $\epsilon$ (the magnitude of $\delta$), as $\delta$ represents the features of the target class. In contrast, *Bad-PFL* is relatively less sensitive to variations in $\sigma$ (the magnitude of $\xi$), since $\xi$ primarily serves to induce misclassification rather than explicitly directing the sample towards a specific class. Nonetheless, $\xi$ remains essential; as $\sigma$ decreases, the performance of *Bad-PFL* also gradually diminishes, although not as dramatically as when $\epsilon$ decreases.

Table 18: The impact of $\epsilon$ and $\sigma$ in attack performance of *Bad-PFL*.

| $\epsilon$ | FedBN | | FedRep | | $\sigma$ | FedBN | | FedRep | |
|---|---|---|---|---|---|---|---|---|---|
| | Acc | ASR | Acc | ASR | | Acc | ASR | Acc | ASR |
| 0 | 80.78 | 13.74 | 80.17 | 12.82 | 0 | 80.56 | 68.56 | 80.20 | 79.32 |
| 1 | 80.70 | 23.05 | 80.32 | 54.92 | 1 | 80.75 | 72.84 | 80.45 | 85.38 |
| 2 | 80.86 | 78.91 | 80.20 | 79.68 | 2 | 80.68 | 77.37 | 80.52 | 88.88 |
| 3 | 80.77 | 80.04 | 80.35 | 86.79 | 3 | 80.84 | 80.52 | 80.23 | 93.86 |
| 4 | 80.72 | 82.22 | 80.29 | 97.95 | 4 | 80.72 | 82.22 | 80.29 | 97.95 |

### B.2.5 CONVERGENCE COMPARISON

Figure 6 illustrates the accuracy of the personalized models and their ASRs using three different attack methods over the training rounds. Overall, when the training round reaches 500, both the models and the attack methods converge.

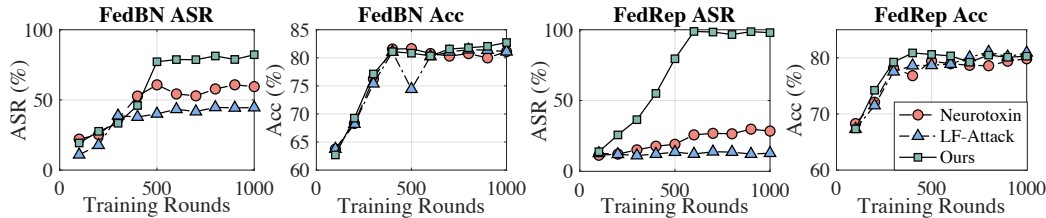

Figure 6: Convergence comparison with various attacks for FedBN and FedRep.

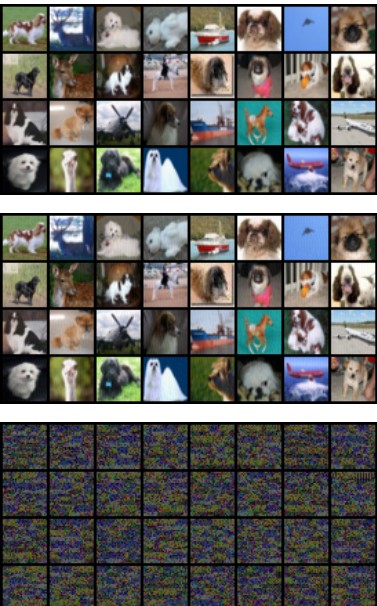

Figure 7: The original images, trigger-added images, and corresponding triggers (amplified by a factor of 20) with $\epsilon = 1$ and $\sigma = 4$.

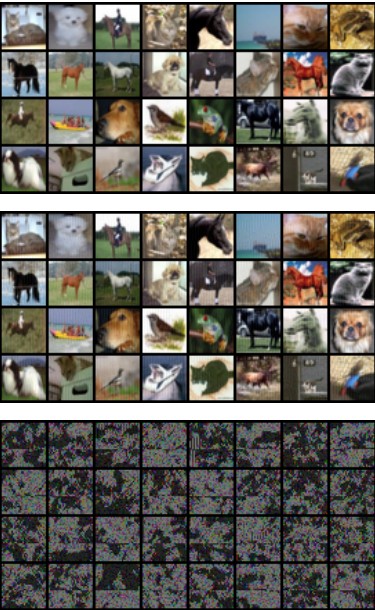

Figure 8: The original images, trigger-added images, and corresponding triggers (amplified by a factor of 20) with $\epsilon = 2$ and $\sigma = 4$.

### B.2.6 TRIGGER VISUALIZATION

To demonstrate the invisibility of our trigger, Figures 7 ∼ 12 visualize the original data, our trigger, and trigger-adding data under different magnitude constraints (over $\frac{1}{255}$, $\frac{2}{255}$, $\frac{3}{255}$, and $\frac{4}{255}$ ) of $\delta$ and $\xi$. The top images represent the original data, the middle images show trigger-added data, and the bottom images display the trigger $(\delta + \xi)$ generated by *Bad-PFL* for each data point. Since the strength of $\delta + \xi$ is quite small, the intensity of the trigger is amplified by a factor of 20, i.e., $20 \times (\delta + \xi)$, for better visualization. First, it is difficult for the human eye to discern any differences

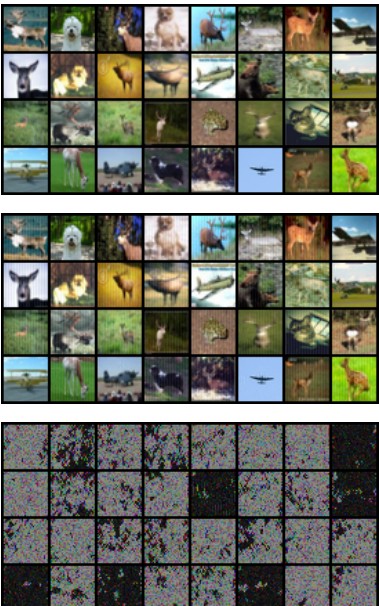

Figure 9: The original images, trigger-added images, and corresponding triggers (amplified by a factor of 20) with $\epsilon = 3$ and $\sigma = 4$.

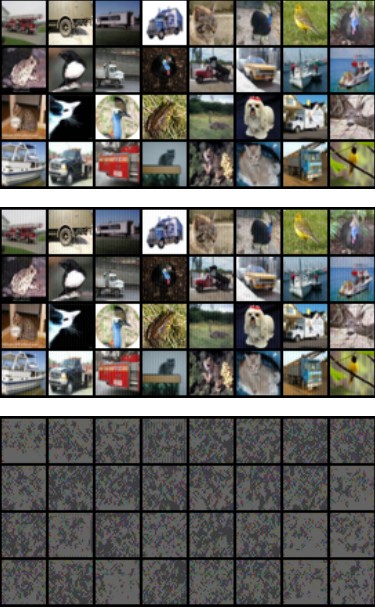

Figure 10: The original images, trigger-added images, and corresponding triggers (amplified by a factor of 20) with $\epsilon = 4$ and $\sigma = 1$.

between the original images and the trigger-added data, often perceiving them as identical, which makes our trigger highly stealthy. Furthermore, it can be seen that *Bad-PFL* generates different trigger patterns for different images, which further enhances the stealthiness.

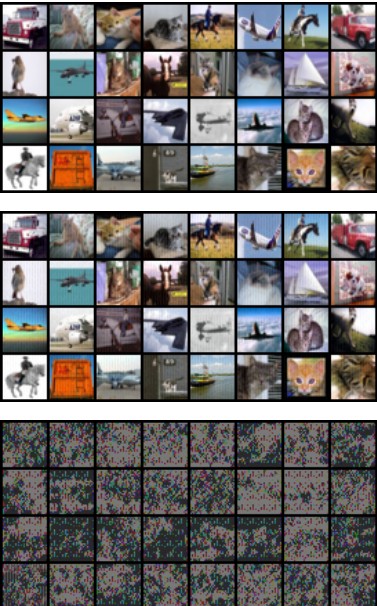

Figure 11: The original images, trigger-added images, and corresponding triggers (amplified by a factor of 20) with $\epsilon = 4$ and $\sigma = 2$.

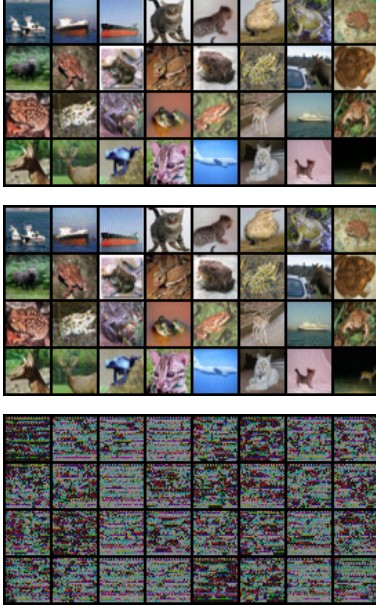

Figure 12: The original images, trigger-added images, and corresponding triggers (amplified by a factor of 20) with $\epsilon = 4$ and $\sigma = 3$.

### B.2.7 THE IMPACT OF CLIENT NUMBER

We here evaluate the impact of the number of clients on the performance of *Bad-PFL*, with the results reported in Table 19. Wherein, we fix the number of malicious clients at 10. We observe that as the total number of clients increases, both accuracy and ASR gradually decline. Though this, *Bad-PFL* still achieves significant ASRs ($> 80\%$). Moreover, when the ratio of compromised clients is fixed at 10%, we see that the performance of *Bad-PFL* remains largely unchanged.

Table 19: The attack performance of *Bad-PFL* against FedRep under varying client numbers. The fixed number setting indicates that the number of compromised clients remains constant at 10, regardless of changes in the total number of clients. The fixed ratio setting specifies that the number of compromised clients corresponds to 10% of the total number of clients.

| Strategy | Fixed Number | | Fixed Ratio | |
|---|---|---|---|---|
| Client Number | Acc | ASR | Acc | ASR |
| 50 | 80.72 | 99.22 | 80.65 | 98.00 |
| 100 | 80.29 | 97.95 | 80.24 | 97.68 |
| 150 | 79.75 | 93.66 | 79.47 | 97.30 |
| 200 | 78.04 | 85.87 | 77.95 | 98.48 |

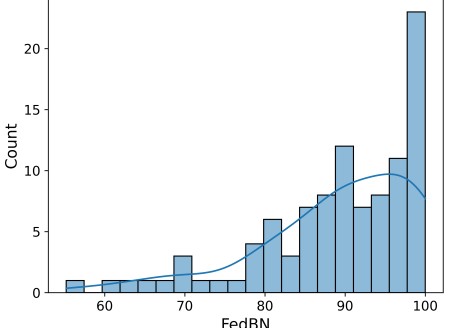 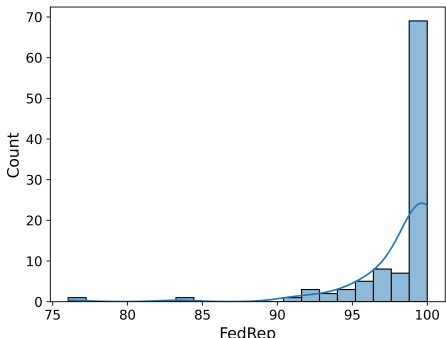

Figure 13: The distribution of ASRs for FedBN and FedRep. For FedBN, the 25th, 50th, and 75th percentiles of ASRs are 84%, 90%, and 96%, respectively. For FedRep, the 25th, 50th, and 75th percentiles are 97%, 100%, and 100%.

### B.2.8 THE DISTRIBUTION OF ASRS

Figure 13 visualizes the ASRs of *Bad-PFL* across different clients using their local test sets. Overall, the ASRs remain high for the majority of clients. In the case of FedRep, *Bad-PFL* achieves ASRs below 90% for only two clients, while the ASRs appear to be more dispersed for FedBN. FedBN allows local personalized models to learn private batch normalization layers tailored to their specific datasets. Therefore, for FedBN, we hypothesize that the models of compromised clients and benign clients learn different feature distributions. Our generator focuses on learning the natural features related to the target label from the models of compromised clients. These features may overlap less with those learned by the models of benign clients, leading to lower ASRs. In contrast, FedRep requires all local personalized models to share a feature extractor and thus enjoys more feature overlap and is more vulnerable to *Bad-PFL*.

Table 20: The performance of *Bad-PFL* in single-target and multi-target attack scenarios.

| Attack | Single-target Attack | | Multi-target Attack | |
|---|---|---|---|---|
| PFL | Acc | ASR | Acc | ASR |
| FedBN | 80.72 | 82.22 | 79.65 | 80.44 |
| FedRep | 80.29 | 97.95 | 78.79 | 96.43 |

### B.2.9 MULTI-TARGET ATTACK

Table 20 reports the performance of *Bad-PFL* in multi-target attack scenario on CIFAR-10. We train a separate generator for each category. As observed, the average performance of the personalized models decreases in multi-target attack scenario compared to single-target attack scenario. This reduction occurs because the global model needs to fit multiple generators simultaneously, which can somewhat impede the learning of the primary task to some extent. Moreover, we see that the ASRs do not appear to be significantly affected.

Table 21: The performance of different attack methods against three defenses. We employ FedRep.

| Attack | Simple-Tuning | | BAERASER | | MAD | |
|---|---|---|---|---|---|---|
| | Acc | ASR | Acc | ASR | Acc | ASR |
| Neurotoxin | **82.65** | 19.80 | **79.59** | 13.05 | 74.52 | 19.46 |
| LF-Attack | 81.68 | 12.59 | 77.90 | 15.24 | **74.81** | 10.49 |
| Perdoor | 81.59 | 63.15 | 79.33 | 84.12 | 74.64 | 46.90 |
| Iba | 81.82 | 49.31 | 77.74 | 78.98 | 74.55 | 55.58 |
| BadFL | 82.24 | 22.79 | 79.39 | 17.59 | 74.75 | 24.73 |
| PFedBA | 81.27 | 42.36 | 78.59 | 31.88 | 74.29 | 55.92 |
| *Bad-PFL* | 82.05 | **88.82** | 77.68 | **91.54** | 74.37 | **90.74** |

### B.2.10 THE COMPARISON OF *Bad-PFL* WITH VARIOUS BACKDOOR ATTACKS ACROSS MULTIPLE DEFENSES

We compare *Bad-PFL* with more backdoor attacks across various defenses. Simple-Tuning (Qin et al., 2023) and BAERASER (Liu et al., 2022) are two post-training defenses. In Simple-Tuning, clients reinitialize their classification heads and train on their local datasets. BAERASER attempts to reverse triggers and then employs forgetting techniques to reduce the model's memory of the recovered triggers. Multi-metrics Adaptive Defense (MAD) (Huang et al., 2023) is a defense method applied during training, integrating multiple metrics to better identify malicious clients. Additionally, we introduce four new attack methods: Perdoor (Alam et al., 2022), Iba (Nguyen et al., 2024), BadFL (Ye et al., 2024b), and PFedBA (Lyu et al., 2024). Both Perdoor and Iba focus on customizing triggers for specific samples, but they do not explore performance in non-IID scenarios. BadFL and PFedBA focus on non-IID settings; however, BadFL is overly concentrated on FedRep. PFedBA generates triggers by solving a gradient matching problem, resulting in fixed triggers that are easily detectable.

Table 21 reports the attack performance of these methods against three defense methods, using FedRep to train the models. Overall, *Bad-PFL* significantly outperforms these attacks in terms of ASRs. The ASRs achieved by *Bad-PFL* are considerably higher than those of Iba, as we utilize disruptive noise to enhance the effectiveness of our triggers. Furthermore, we observe that fixed-trigger attacks, such as PFedBA, are easily countered by BAERASER, because fixed triggers are more easily recoverable. In contrast, dynamic-trigger attacks, including Perdoor, Iba, and *Bad-PFL*, demonstrate strong resilience.

### B.2.11 EVALUATION OF BACKDOOR ATTACK STEALTHINESS

We examine the stealthiness of *Bad-PFL* from two perspectives: 1) whether benign clients can detect backdoors in their models, and 2) whether benign clients can recognize trigger-added samples. For the first perspective, we utilize Neural Cleanse, which computes an anomaly index by recovering trigger candidates to convert all clean images to each label. If the anomaly index for a specific label is significantly higher than for others, it indicates that the model is likely compromised. We evaluate different attack methods by calculating the anomaly index for the target label using Neural Cleanse. A smaller anomaly index suggests that the backdoor attack is harder to detect. For the second perspective, we employ STRIP, which identifies trigger-added samples based on the prediction entropy of input samples generated by applying different image patterns. Higher entropy signifies a more stealthy trigger.

We train ResNet10 with FedRep on CIFAR-10. By default, we select the models of the first ten benign clients and the CIFAR-10 test set to estimate the anomaly index and entropy. Table 22 reports the results. The average anomaly index for non-target labels is 1.9, while the entropy of clean samples is 0.92. As expected, *Bad-PFL* achieves a lower anomaly index and higher entropy compared to baseline attacks, demonstrating superior stealthiness.

## C DISCUSSION ON ATTACK COSTS

We here discuss the overhead associated with our attack method, examining both the training and inference phases. During the FL process, *Bad-PFL* involves the optimization of the generator and

Table 22: The performance of different backdoor attack methods against state-of-the-art detection methods. The anomaly index presented here is calculated for the target label, with the best results highlighted in bold.

| Detection Method | Neural Cleanse (Anomaly Index) | STRIP (Entropy) |
|---|---|---|
| Neurotoxin | 5.8 | 0.13 |
| LF-Attack | 5.7 | 0.12 |
| PFedBA | 4.9 | 0.25 |
| *Bad-PFL* | **2.2** | **0.77** |

Table 23: Total time taken (in seconds) for the client to run local training using different attack methods. We follow the training configuration in Section 4.2.

| Attack | FedProx | SCAFFOLD | FedBN | FedRep | Ditto |
|---|---|---|---|---|---|
| No Attack | 0.453 | 0.211 | 0.201 | 0.447 | 0.451 |
| Neurotoxin | 0.475 | 0.223 | 0.213 | 0.452 | 0.468 |
| Perdoor | 5.744 | 3.273 | 3.113 | 3.349 | 3.358 |
| Iba | 0.791 | 0.661 | 0.620 | 1.227 | 1.178 |
| BapFL | 0.982 | 0.578 | 0.552 | 0.797 | 0.552 |
| PFedBA | 1.820 | 1.540 | 1.480 | 1.649 | 1.443 |
| *Bad-PFL* | 0.818 | 0.620 | 0.613 | 1.206 | 1.132 |

the training of the global model on trigger-added data. On the one hand, the optimization of the generator, as described in Equation 7, requires two complete forward and backward passes of the global model, along with one forward and backward pass of the generator. On the other hand, optimizing the global model on trigger-added data involves crafting triggers, which entails a single forward pass of the generator (for $\delta$), as well as a forward and backward pass of the global model (for $\xi$).

Table 23 empirically evaluates the time required for compromised clients to execute local training using various attack methods. We conduct these experiments using CIFAR-10, with the reported times averaged over 100 trials on a single RTX 4090 GPU. "No Attack" indicates the time taken for a client to perform local training without executing backdoor attacks. Table 23 does not report the costs associated with LF-Attack, as it needs training models from scratch multiple times (in a linear relationship with the number of layers in the neural networks) to evaluate each layer's significance for backdoor attacks. The attack costs for LF-Attack are significantly higher than those of existing backdoor attack methods, and we will not discuss it further.

We observe that Neurotoxin incurs the lowest attack overhead since it utilizes a fixed trigger; however, this also results in lower attack performance (as shown in Table 21). More advanced backdoor attack methods often employ more sophisticated trigger generation techniques. For instance, Perdoor uses the BIM method to create triggers, necessitating multiple complete forward and backward passes of the global model (10 times here). PFedBA has to handle a gradient matching problem, requiring at least two forward and backward passes of the global model for each optimization iteration of the trigger. Our attack method also demands a certain amount of time investment. Nevertheless, we stress that compared to existing attack methods, our attack method still achieves superior performance while maintaining a competitive time overhead. Moreover, federated backdoor attack methods focus more on attack performance over runtime costs, as the primary bottleneck in FL lies in communication costs. These attack methods usually require only a few seconds, which is small compared to communication durations, making them less detectable in practice. In the inference phase, our method for generating triggers for 32 data samples takes approximately 0.07 seconds, which is also quite efficient. In summary, *Bad-PFL* is practical.

# D  A CLOSE LOOK AT *Bad-PFL*

We here explain how *Bad-PFL* effectively overcomes the three challenges in Section 2.2. The trigger employed in *Bad-PFL* consists of target feature ($\delta$) and disruptive noise ($\xi$). Naturally, data from the target label inherently contains $\delta$ and the relationship between $\delta$ and the target label (established through human labeling). Recall that we train models to maximize accuracy. Thus, models tend to

leverage any available features to do so. This means that as long as the clients' datasets include data from the target label, personalized models will inevitably learn $\delta$ and the relationship between $\delta$ and the target label. This enables *Bad-PFL* to effectively address the challenges in Section 2.2.

More specifically, in full model-sharing methods, relying solely on the regularization term is inadequate for transferring the backdoor to personalized models. *Bad-PFL* leverages the natural features of the target class as our trigger, which are inherently present in the data associated with that class, including the local datasets of benign clients. Personalized models trained on benign clients' local datasets will actively learn the natural features and the relationship from the natural features to the target label for higher accuracy. The guidance provided by the regularization term also further enhances this learning process, allowing *Bad-PFL* to effectively overcome the first challenge.

In partial model-sharing methods, the challenge lies in effectively conveying the connection between the triggers and the target label to the personalized models. Since we cannot alter the local training processes of benign clients, it is nearly impossible to embed the relationship between handcrafted triggers and the target label through data poisoning or other means. Instead, *Bad-PFL* utilizes the natural features of the target label. This mapping between natural features and the target label, which already exists in the local datasets of benign clients, allows us to effectively address the second challenge without needing to modify the training process of benign clients.

Regarding the dilution of backdoors, we recognize that the clients' datasets contain these natural features and their relationships with the target label. During the fine-tuning or training process, the model is less likely to forget these relationships because doing so would lead to a decline in performance. In other words, the presence of these natural features in the training data reinforces the model's memory of the backdoor, mitigating the risk of it being overwritten or lost. In summary, the above analysis clearly illustrates how *Bad-PFL* successfully overcomes the three challenges previously mentioned.

Importantly, even if a particular client's dataset lacks data from the target label, *Bad-PFL* probably remains effective. First, in practice, only a small number of client datasets may lack target class data, making it unlikely that the global model fails to learn $\delta$ and the mapping from $\delta$ to the target label. Moreover, in *Bad-PFL*, malicious clients actively promote the model's reliance on $\delta$ to predict the target class (Equation 7). The similarity constraint between the global model and the personalized models encourages the personalized models to leverage the relationship between $\delta$ and the target class more effectively. This encourages the personalized models to also utilize the relationship between $\delta$ and the target class to a greater extent. Second, we introduce destructive noise $\xi$, which interferes with features belonging to the true class, thereby allowing $\delta$ to function more effectively in the decision-making process of personalized models. These two unique designs can enhance the performance of *Bad-PFL*. The only conceivable countermeasure would be if clients fine-tune their personalized models without including target class data; however, this absence would significantly degrade their performance on the target class.

**Empirical evidence.** To further substantiate our claims, we present experimental results. First, we demonstrate that $\delta$ utilized in *Bad-PFL* are indeed natural features of the target class. We employ t-SNE to visualize the features extracted from test set of CIFAR-10 by the global model, alongside $\delta$ for these test samples. As illustrated in Figure 14, the model classifies $\delta$ as belonging to the target class, indicating that it recognizes $\delta$ as natural features of the target class.

Next, we validate the effectiveness of the disruptive noise $\xi$. Similarly, we use t-SNE to visualize the features of both the test samples with and without $\xi$. Figure 15 reveals that, while the features from $x$ cluster neatly by class, those from $x + \xi$ exhibit a more chaotic distribution. This confirms that $\xi$ effectively disrupts the features associated with their ground-truth classes.

We also conduct numerical experiments to further substantiate our conclusions. We train a ResNet10 on the CIFAR-10 dataset from scratch using three distinct configurations. The first configuration employs a standard training setup. In the second configuration, we add disruptive noise $\xi$ to the training samples of the target label during each iteration. Building on the second configuration, the third configuration introduces $\delta$ into the training samples of the target label. Intuitively, the disruptive noise is expected to corrupt the features of the training samples of the target label, which would hinder the model from learning the underlying features of the target label, resulting in poor performance on those samples. In the third configuration, if $\delta$ accurately captures the features of the target label, we anticipate that the model will learn more about the target label compared to the

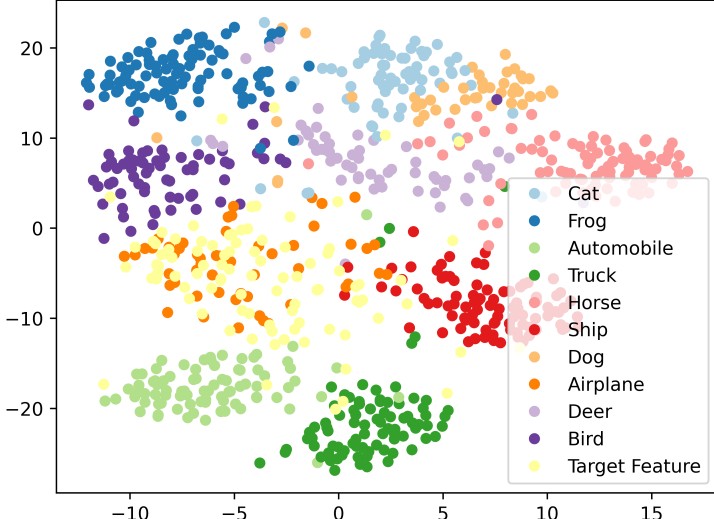

Figure 14: We visualize the features from the fully connected layer of the CIFAR-10 test set. More-over, we feed the test samples into the generator $\mathcal{G}_w$ to produce the corresponding $\delta$. For better visualization, we scale the norm of $\delta$ to match the magnitude of the test samples' norm. As can be seen, the features of generated $\delta$ significantly overlap with the features of the target class (Airplane). This suggests that $\delta$ indeed represents the natural features of the target label.

Table 24: We train ResNet10 using three different configurations. The first one is the standard setup. The second one introduces disruptive noise $\xi$ to the training samples of the target label during each iteration. Building on this second configuration, the third one adds $\delta$ into the training samples of the target label. We report the accuracy of the trained models on the CIFAR-10 test set, as well as specifically on the test samples from the target label.

| Setup | Acc | Acc of the Target Label |
|---|---|---|
| First Configuration (Standard Training) | 80.70 | 80.30 |
| Second Configuration (with $\xi$) | 70.70 | 6.70 |
| Third Configuration (with $\delta + \xi$)) | 72.90 | 39.10 |

second configuration, leading to better performance in the samples of the target label. We reuse the generator in Section 4.2 of the original manuscript (against FedRep).

Table 24 reports the accuracy of the model on the entire test set of CIFAR-10, as well as on the test samples from the target label alone. We observe that the model achieves an accuracy of only 6.70% on the samples of the target label, indicating that the disruptive noise indeed significantly impairs the features of the samples of the target label. In the third configuration, we see that the model's accuracy on the samples of the target label rebound to 39.10%. This suggests that our generator indeed learns the features of the target label.

# E  THE INTERPLAY BETWEEN $\delta$ AND $\xi$

We here study the interplay between $\delta$ and $\xi$. In detail, we evaluate the proportion of pixels where $\delta$ and $\xi$ share the same sign, finding it to be approximately 26.28%, averaged over 1000 samples. This indicates that $\delta$ and $\xi$ do not completely align in terms of the direction of pixel changes, suggesting a more intricate interplay between $\delta$ and $\xi$. To further clarify the relationship between the $\delta$ and $\xi$, we have included visualizations of $\xi$ and $\delta + \xi$ to better illustrate their effects on pixel value changes. As illustrated in Figure 16, the pixel changes introduced by $\xi$ appear somewhat erratic from a human perspective. In contrast, the combined effect of $\xi + \delta$ exhibits a clear pattern, predominantly altering pixels in the upper right corner. This highlights the interplay between $\delta$ and $\xi$, characterized by both resistance and agreement. While $\xi$ proposes specific pixel change directions, $\delta$ can either amplify

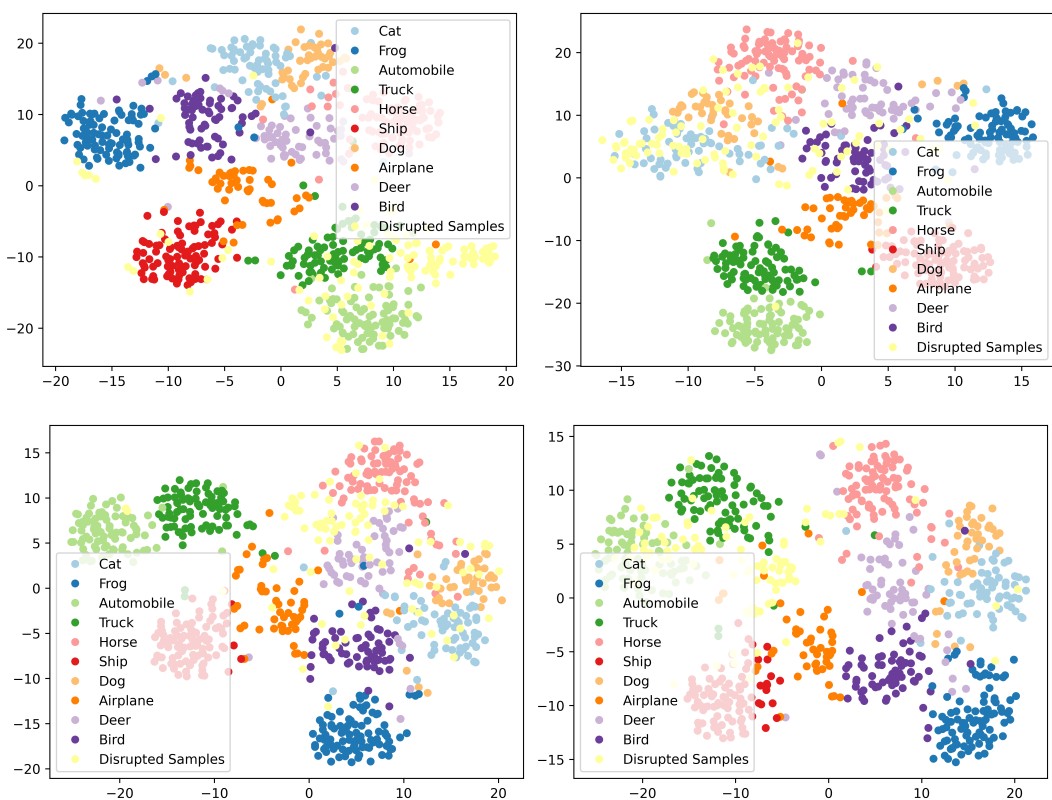

Figure 15: We visualize the features of $x$ and $x + \xi$ from the fully connected layer using t-SNE. Here, $x$ comes from the test set of CIFAR-10. We extract images of the Automobile, Dog, Horse, and Truck categories from the test set and subsequently craft $\xi$ for each. The top left corresponds to Automobile, the top right to Dog, the bottom left to Horse, and the bottom right to Truck. We see that the features of $x + \xi$ indeed deviate significantly from the target cluster, demonstrating that $\xi$ effectively disrupts the features associated with their corresponding ground-truth classes.

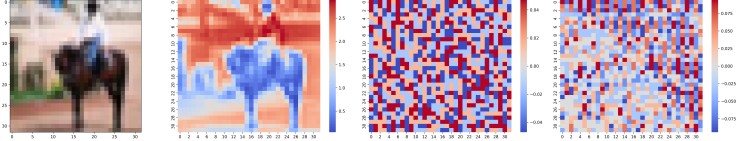

Figure 16: Case 1. The leftmost one is the original image. The second image is a heatmap generated by summing the pixel values across different channels of this image. The third and fourth images represent the heatmaps obtained by summing the channels of $\xi$ and $\xi + \delta$, respectively.

or counteract these suggestions. This means that $\delta + \xi$ reflects a negotiation between the two: $\delta$ may dampen or redirect some of the changes suggested by $\xi$. This dynamic can lead to concentrated perturbations in certain areas of the input, indicating that $\xi$ selectively agrees with the changes

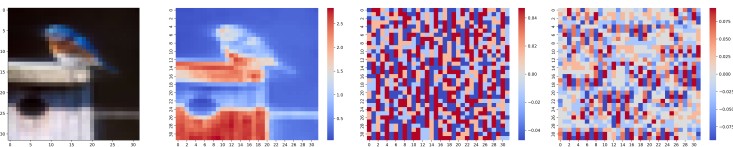

Figure 17: Case 2.

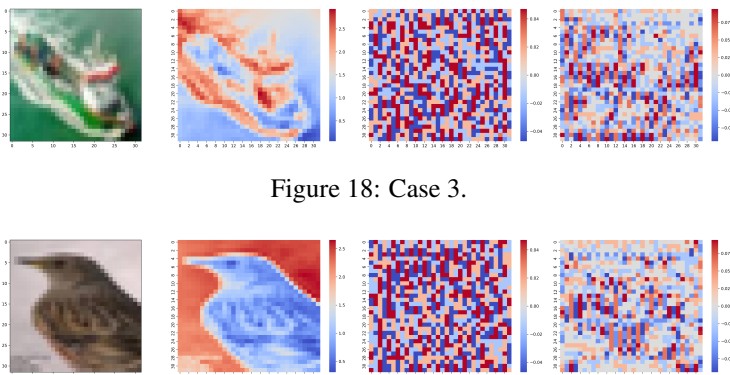

Figure 18: Case 3.

Figure 19: Case 4.

proposed by $\delta$. This phenomenon can be observed in Figures 17, 18, and 19, reinforcing the notion that there exists a complex interaction between $\xi$ and $\delta$, rather than a straightforward combination into a single effect.

