# OpenReview forum: "Bad-PFL: Exploiting Backdoor Attacks against Personalized Federated Learning"
_ICLR.cc/2025/Conference — ICLR 2025 Poster_

### Official Review · Reviewer_easG · 2024-11-03

**Soundness:** 3
**Presentation:** 3
**Contribution:** 2
**Rating:** 6
**Confidence:** 3

**Summary:**

The authors investigate the potential of backdoor attacks targeting personalized federated learning (PFL) and intuitively outline why such attacks often fail in PFL. They propose using natural features rather than manually designed triggers to execute backdoor attacks in PFL, which improves both the attack's success rate and robustness across various test cases, with or without defenses.

**Strengths:**

1. Clear and well-structured presentation
2. Comprehensive experimental test cases covering various attacks, defenses, and PFL mechanisms
3. Novel reinforcement techniques, including generating target features

**Weaknesses:**

1. Fundamentally, I feel like the traditional backdoor objective may be insufficient for effectively attacking personalized federated learning (PFL), especially in scenarios with a high degree of non-i.i.d. data where each client has its own unique "target." Therefore, a personalized or adaptive backdoor attack approach may be necessary.
2. The current attack mechanism is based on empirical intuitions. Strengthening this work could involve a deeper exploration of the model’s internal dynamics, such as identifying which layers or neurons are critical for shared vs. unshared parameters and examining the influence of regularization on the model. Alternatively, providing a fundamental analysis of why traditional backdoor attacks fail and why Bad-PFL succeeds would also add significant value.
3. As this is the first study on backdoor attacks in PFL, rather than focusing on more practical backdoor scenarios, it’s recommended to compare white-box and black-box attacks. This comparison can help illustrate the limitations of current attacks, especially when the attacker has knowledge of clients' data and the full model structure, thus providing justification for the authors' three hypotheses on why the backdoor was ineffective in PFL.

**Questions:**

1. The authors consider using natural features as triggers, termed an edge-case attack in [1]. I wonder if the main distinction of the proposed attack is that it generates target features rather than manually selecting them. However, it seems that different clients—or at least different client groups—may require distinct natural features for effectiveness.
2. Other than training-stage defenses (i.e., robust aggregation), have the authors considered post-training defenses (e.g., pruning, CRFL [2]) or unlearning as defenses? I feel those defenses may be more effective on backdoor attacks.
3. For different clients, does the effectiveness of the backdoor vary significantly? And is this variation caused by data heterogeneity or differences in shared model parameters?
4. What happens when attackers target different labels and clients have multiple objectives? Detailed experiments may not be necessary, but insights and intuitions would be helpful.

[1] Wang, Hongyi, et al. "Attack of the tails: Yes, you really can backdoor federated learning." Advances in Neural Information Processing Systems 33 (2020): 16070-16084.
[2] Xie, Chulin, et al. "Crfl: Certifiably robust federated learning against backdoor attacks." International Conference on Machine Learning. PMLR, 2021.

---

> ### Author Response · Authors · 2024-11-17
> **Response (1/3)**
>
> We appreciate the time and effort you invested in reviewing this manuscript. Your insightful comments, particularly regarding Q3, have significantly enhanced the quality of this paper. We have uploaded the revised manuscript, and the changes are highlighted in red for your convenience. Specifically, the experimental results related to Q4 and Q5 have been included in Appendix B.2.10 and Appendix B.2.9, respectively. Below, we provide a detailed response to each of your comments. Unless otherwise stated, the experimental configuration defaults to that used in Section 4.2.
>
> ---
>
> **Q1:** The current attack mechanism is based on empirical intuitions. Strengthening this work could involve a deeper exploration of the model’s internal dynamics, such as identifying which layers or neurons are critical for shared vs. unshared parameters and examining the influence of regularization on the model. Alternatively, providing a fundamental analysis of why traditional backdoor attacks fail and why Bad-PFL succeeds would also add significant value.
>
> **Response:**
> In fact, partial model-sharing methods focus on determining which parameters or layers within the model should be shared. The prevailing consensus is that the early layers of the model should be shared, while the later layers should remain private to the clients. However, some literature presents differing opinions, such as FedBN, which advocates for the privatization of intermediate layers, specifically batch normalization layers. Currently, there is no definitive agreement on this issue.
>
> Regarding the impact of regularization, we discuss this in Section 2.2. Specifically, regularization terms tend to align parameters that significantly affect the performance of personalized models on local data. Based on these insights, we can improve our attack method by extracting natural features associated with more prominent parameters or layers. We plan to explore this in future work.
>
> Additionally, we have included a discussion in Appendix D that explains how our attack method can address the challenges mentioned in Section 2. Please refer to Appendix D for further details.
>
>
> ---
>
> **Q2:** As this is the first study on backdoor attacks in PFL, rather than focusing on more practical backdoor scenarios, it’s recommended to compare white-box and black-box attacks. This comparison can help illustrate the limitations of current attacks, especially when the attacker has knowledge of clients' data and the full model structure, thus providing justification for the authors' three hypotheses on why the backdoor was ineffective in PFL.
>
> **Response:** Thanks for your comment. In fact, the experiments conducted in Section 2.2 consider a white-box scenario. The measures proposed there require attackers to modify the federated learning training settings, such as employing weighted regularization for existing backdoor attack methods. We acknowledge that we did not take into account the attacker’s knowledge of the client data distribution. We plan to explore how this knowledge influences the design of backdoor attacks in future work. Additionally, we believe that considering a practical backdoor scenario is also of significant importance.
>
> ---

---

> ### Author Response · Authors · 2024-11-17
> **Response (2/3)**
>
> **Q3:** The authors consider using natural features as triggers, termed an edge-case attack. I wonder if the main distinction of the proposed attack is that it generates target features rather than manually selecting them. However, it seems that different clients—or at least different client groups—may require distinct natural features for effectiveness.
>
> **Response:**
> This is a very insightful comment. edge-case attack [7] changes the label of edge-case samples (that are located in the tail of the input distribution) to trick the model into classifying edge-case samples as the target label. The core idea is that edge-case samples have distinct features compared to non-edge-case samples, and the model's high capacity allows it to fit these differences. If edge-case samples are rare across most clients and only appear in malicious clients, the model tends to believe that the features of these edge-case samples correspond to the target label.
>
> Intuitively, our attack method indeed focuses on generating target features rather than manually selecting them. However, there are several key differences. First, edge-case attack aims to encourage the model to associate the features of edge-case samples with the target label, regardless of whether those edge-case samples belong to the target label. This means the model might also link features of non-target labels to the target label. In contrast, our attack method utilizes a generator to extract features specifically learned by the model for the target label, ensuring that the natural features are primarily associated with the target label, with little relation to non-target labels. This is a significant distinction. Second, as you pointed out, we leverage the generator to generate target features rather than relying on manual selection, which typically results in better performance while reducing the costs and biases associated with manual selection. Third, our attack can induce targeted misclassifications for any sample, not just limited to edge-case samples.
>
> Regarding your final question, you are correct; this pertains to the non-IID problem. As heterogeneity increases, models across different clients must learn distinct features to accommodate their local data distributions. Consequently, the divergence between the optimal models for different clients becomes more pronounced with rising heterogeneity, making backdoor attacks more challenging to execute. Our experimental results (Figure 5) support this observation, demonstrating that as data heterogeneity increases, the ASRs of federated backdoor attacks against PFL methods decline.
>
> In fact, the ASRs of our attack method on a small number of clients are not very high (See response to Q5 or Appendix B.2.8). This is especially pronounced in FedBN, which allows personalized models to learn private batch normalization layers tailored to their local datasets. We observe that the ASRs for a minority of clients are around 60\%, which we suspect is due to the need for these clients' models to learn different group features. In contrast, FedRep fixes the feature extractor, requiring all personalized models to learn the same underlying features. For FedRep, we see little variation in ASRs across different clients.
>
> ---
>
> **Q4:** Other than training-stage defenses, have the authors considered post-training defenses or unlearning as defenses? I feel those defenses may be more effective on backdoor attacks.
>
> **Response:**
> We have included the latest defense methods [5, 6], with specific results reported in the table below. We employ FedRep, and this evaluation has been included in the revised paper (Appendix B.2.10). Notably, Simple-Tuning and BAERASER are two state-of-the-art post-hoc defense methods. BAERASER employs unlearning techniques to eliminate the model's memory of triggers. Furthermore, based on suggestions from other reviewers, we have also added four advanced backdoor attack methods, including Perdoor, Iba, BapFL, and PFedBA, and a defense, MAD. Overall, Bad-PFL demonstrates a significant advantage over these attacks in terms of ASRs against three defenses. For more discussions see Appendix B.2.10.
>
> |   Defense  | Simple-Tuning [5] |        | BAERASER [6] |        |   MAD [7]  |        |
> |:----------:|:-------------:|:------:|:--------:|:------:|:------:|:------:|
> |   Attack   |      Acc      |   ASR  |    Acc   |   ASR  |   Acc  |   ASR  |
> | Neurotoxin |     82.65     | 19.80  |  78.59   | 13.05  | 74.52  | 19.46  |
> |  LF-Attack |     81.68     | 12.59  |  77.90   | 15.24  | 74.81  | 10.49  |
> |   Perdoor [1]  |     81.59     | 63.15  |  79.33   | 84.12  | 74.64  | 46.90  |
> |     Iba [2]     |     81.82     | 49.31  |  77.74   | 78.98  | 74.55  | 55.58  |
> |    BapFL [3]   |     82.24     | 22.79  |  79.39   | 17.59  | 74.75  | 24.73  |
> |   PFedBA [4]   |     81.27     | 42.36  |  78.59   | 31.88  | 74.29  | 55.92  |
> |     Our    |     82.05     | 88.82  |  77.68   | 91.54  | 74.37  | 90.74  |

---

> > ### Author Response · Authors · 2024-11-17
> > **Response (3/3)**
> >
> > **Q5:** For different clients, does the effectiveness of the backdoor vary significantly? And is this variation caused by data heterogeneity or differences in shared model parameters?
> >
> > **Response:** We visualize the ASRs of our attack method across different clients using their local test sets. The figure can be found in Appendix B.2.8. Overall, the ASRs remain high for the majority of clients. Specifically, for FedBN, the 25th, 50th, and 75th percentiles of ASRs are 84\%, 90\%, and 96\%, respectively. For FedRep, the 25th, 50th, and 75th percentiles are 97\%, 100\%, and 100\%. Notably, for FedRep, we observe ASRs of 76\% and 84\% for two specific clients. We believe this variation is primarily due to data heterogeneity, which we discuss in response to the previous question, so we will not elaborate further here. As for the higher ASRs in FedRep, we attribute this to the requirement that all personalized models share the same feature extractor.
> >
> > ---
> >
> > **Q6:** What happens when attackers target different labels and clients have multiple objectives? Detailed experiments may not be necessary, but insights and intuitions would be helpful.
> >
> > **Response:**
> > We now consider multi-target attack scenario. In this scenario, attackers design several distinct triggers, each aimed at manipulating the model to misclassify inputs as different desired labels. In our attack, this means training a separate generator for each label to produce natural features corresponding to that label. We evaluate our attack method on CIFAR-10, with results reported in the table below. As noted, the average performance of the personalized models declines in the multi-target attack scenario compared to the single-target attack scenario. This decrease occurs because the global model has to accommodate multiple generators simultaneously, which can somewhat hinder the learning of the primary task. Furthermore, it appears that the ASRs remain largely unaffected.
> >
> > | Attack | Single-target Attack |       | Multi-target Attack |       |
> > |:------:|:--------------------:|:-----:|:-------------------:|:-----:|
> > |   PFL  |          Acc         |  ASR  |         Acc         |  ASR  |
> > |  FedBN |         80.72        | 82.22 |        79.65        | 80.44 |
> > | FedRep |         80.29        | 97.95 |        78.79        | 96.43 |
> >
> > ---
> >
> > Reference:
> >
> > [1] Perdoor: Persistent non-uniform backdoors in federated learning using adversarial perturbations
> >
> > [2] Iba: Towards irreversible backdoor attacks in federated learning
> >
> > [3] Bapfl: You can backdoor personalized federated learning
> >
> > [4] Lurking in the shadows: Unveiling stealthy backdoor attacks against personalized federated learning
> >
> > [5] Revisiting personalized federated learning: Robustness against backdoor attacks
> >
> > [6] Backdoor defense with machine unlearning
> >
> > [7] Attack of the tails: Yes, you really can backdoor federated learning.

---

> ### Author Response · Authors · 2024-11-19
> **Discussion Inquiry**
>
> Dear Reviewer easG,
>
> We thank you for the precious review time and valuable comments. We have provided responses to your questions and the weakness you mentioned. We hope this can address your concerns.
>
> We would appreciate the opportunity to discuss whether your concerns have been addressed appropriately. Please let us know if you have any further questions or comments. We look forward to hearing from you soon.
>
> Best regards,
>
> Authors

---

> > ### Author Response · Authors · 2024-11-22
> > **Looking forward to your feedback**
> >
> > Dear Reviewer easG,
> >
> > Sorry to bother you again. With the discussion phase nearing the end, we would like to know whether the responses have addressed your concerns.
> >
> > Should this be the case, we are encouraged that you raise the final rating to reflect this.
> >
> > If there are any remaining concerns, please let us know. We are more than willing to engage in further discussion and address any remaining concerns to the best of our abilities.
> >
> > We are looking forward to your reply. Thank you for your efforts in this manuscript.
> >
> > Best regards,
> >
> > Authors

---

### Official Review · Reviewer_mG9G · 2024-11-04

**Soundness:** 2
**Presentation:** 3
**Contribution:** 3
**Rating:** 6
**Confidence:** 5

**Summary:**

This paper investigates backdoor attacks in FPL and introduces a new method to conduct attacks with PFL, named Bad-PFL. This method trains a sample-specific generator to generate a mask for input. Another mask is learned for the main task; the final poisoned sample combines the original one and these two masks. Experiments show that this method can work with different settings and bypass existing defenses.

**Strengths:**

- It provides an interesting insight into why existing backdoor attacks are ineffective in PFL, offering a foundation for advancing attack strategies.
- The proposed Bad-PFL approach achieves high stealth and persistence in personalized models under various settings.

**Weaknesses:**

1. This method shows degradation when the data is highly heterogeneity, which is more practical in cases in FPL.
2. The using of a generator for mask optimization is not novel, previous works such as IBA[1] and Perdoor[2] use the same method. However, this paper lacks comparison and discussion with these papers.
3. The using of two masks now is not totally convincing and lacks of theoretical analysis to support this mechanism. What is exactly the role of each mask in the overall method and what is the relationship of these two masks? The **paper may not fully justify why two separate masks are essential** rather than using a single mask that can adaptively balance target and non-target features.  How can we ensure that the two masks—the target feature enhancement mask ($\delta$) and the disruptive noise mask ($\xi$)—do not interfere with each other, either by collapsing into a single effect or by unintentionally complementing each other? Table 4 is not enough for this point.
4. It is unclear how Bad-PFL can address the limitations mentioned in Section 2.2. Though the authors mentioned using features of the target label as the trigger for a more effective backdoor attack, the benign mask $\eta$ is learned as a noise mask instead of the true features in the images. Could the authors argue this point?

[1] Nguyen, Thuy Dung, et al. "Iba: Towards irreversible backdoor attacks in federated learning." *Advances in Neural Information Processing Systems* 36 (2024).

[2] Alam, Manaar, Esha Sarkar, and Michail Maniatakos. "Perdoor: Persistent non-uniform backdoors in federated learning using adversarial perturbations." *arXiv preprint arXiv:2205.13523* (2022).

**Questions:**

1. In the threat model (Line 236-237), the attack can be colluded or non-colluded but the paper did not explicitly discuss or show this attack can work with both cases.
2. Can the authors explain more detail about the phenomenon mentioned in L263-264, and why the model should focus on the backdoor mask $\delta$?
3. Could the authors discuss the computational overhead of this method? Will it raise suspicion when it takes too long for the malicious to complete a training iteration?
4. In Line 294, what is the intuition of setting \eta by the reversed sign of the input?
Why does it ensure to return of an approximate solution?
5. Will another factor such as the number of clients affect the performance of this method, since it will affect the global model in each training round?
6. Some close related papers are missing such as [3][4]. The authors can discuss the key difference in terms of the methodology of this work.
7. Can the authors elaborate more on the sharp difference in ASR among different datasets such as CIFAR100 and SVHN under FedREP in Tables 15 and 16?

[3] Ye, Tiandi, et al. "BapFL: You can Backdoor Personalized Federated Learning." *ACM Transactions on Knowledge Discovery from Data* (2024).

[4] Lyu, Xiaoting, et al. "Lurking in the shadows: Unveiling Stealthy Backdoor Attacks against Personalized Federated Learning." *arXiv preprint arXiv:2406.06207* (2024).

---

> ### Author Response · Authors · 2024-11-17
> **Response (1/4)**
>
> We have made revisions in response to your constructive comments, which have greatly improved the quality of this paper. The revised manuscript has been uploaded, with changes highlighted in red for your convenience. Specifically, revisions related to Q2 & Q10 can be found in Appendix B.2.10. Revisions for Q3 and Q9 are also included in Appendix D and Appendix B.2.7. Additionally, there are several minor revisions that we have not listed individually. Below, we provide a detailed response to each of your comments. Unless otherwise specified, the experimental configuration defaults to that described in Section 4.2.
>
> ---
>
> **Q1:** This method shows degradation when the data is highly heterogeneity, which is more practical in cases in FPL.
>
> **Response:** As the degree of data heterogeneity increases, the differences between the optimal models of various clients also become more pronounced, which in turn complicates the effectiveness of backdoor attacks. Thus, It has to acknowledge that regardless of the backdoor attack method employed, an increase in heterogeneity will inevitably lead to a decline in attack performance, as illustrated in Figure 5.
>
> Moreover, we would like to clarify that under the same degree of data heterogeneity, our attack method consistently achieves higher ASRs, as shown in Figure 5. For instance, when alpha is set to 0.05, as shown in Table 6, a client’s dataset may predominantly consist of data from only two classes, representing a scenario of significant data heterogeneity. In this case, our attack method still manages to achieve an ASR of about 75% (Figure 5, FedBN). This number is significant and indicates that our attack performance is far from inadequate. Therefore, it is hard to assert that our attack performance is not high.
>
> ---
>
> **Q2:** The using of a generator for mask optimization is not novel, previous works such as IBA and Perdoor use the same method. However, this paper lacks comparison and discussion with these papers.
>
> **Response:**
> IBA [2] employs a generator to produce a noise as a trigger for a given sample. However, there is a key difference in terms of trigger generation. Our triggers incorporate disruptive noise, which effectively directs the model's attention towards the target feature $\delta$. Moreover, when optimizing the client's local model, our poisoning objective aims to maximize the probability of classifying $x+\delta+\xi$ into the target label. The inclusion of $\xi$ corrupts the features of $x$ associated with the true label, enabling the model to better learn the relationship between $\delta$ and the target label. Similarly, during the training of the generator, the presence of $\xi$ aids in improving the generator's ability to learn the target features.
>
> Perdoor [1] does not utilize a generator to produce triggers; instead, they construct triggers based on key parameters within the model. In summary, our technical approach is fundamentally distinct from those in IBA and Perdoor.
>
> Moreover, we would like to highlight the novelty of this manuscript. In the scientific community, novelty is more about contributing new insights to a field rather than merely comparing superficial similarities in techniques [8]. Unlike IBA and Perdoor, we investigate the issue of backdoor attacks in personalized federated learning and clarify why traditional backdoor methods fail in this context. We explain how triggers generated by a generator (target feature) are more effective in this setting and demonstrate how disruptive noise enhances attack performance. To the best of our knowledge, these have not been found in existing research, underscoring the novelty of this manuscript.
>
> We have also conducted evaluations to compare our attack with the works you mentioned (Q2 & Q10), with the attack results reported in the below table. In line with suggestions from other reviewers, we have also added three backdoor defenses: Simple-Tuning, BAERASER, and MAD. We employ FedRep. Overall, Bad-PFL shows a significant advantage over these attacks in terms of ASRs when evaluated against three different defenses. More detailed discussions can be found in Appendix B.2.10.
>
> |   Defense  | Simple-Tuning [5] |        | BAERASER [6] |        |   MAD [7]  |        |
> |:----------:|:-------------:|:------:|:--------:|:------:|:------:|:------:|
> |   Attack   |      Acc      |   ASR  |    Acc   |   ASR  |   Acc  |   ASR  |
> | Neurotoxin |     82.65     | 19.80  |  78.59   | 13.05  | 74.52  | 19.46  |
> |  LF-Attack |     81.68     | 12.59  |  77.90   | 15.24  | 74.81  | 10.49  |
> |   Perdoor [1]  |     81.59     | 63.15  |  79.33   | 84.12  | 74.64  | 46.90  |
> |     Iba [2]     |     81.82     | 49.31  |  77.74   | 78.98  | 74.55  | 55.58  |
> |    BapFL [3]   |     82.24     | 22.79  |  79.39   | 17.59  | 74.75  | 24.73  |
> |   PFedBA [4]   |     81.27     | 42.36  |  78.59   | 31.88  | 74.29  | 55.92  |
> |     Our    |     82.05     | 88.82  |  77.68   | 91.54  | 74.37  | 90.74  |

---

> ### Author Response · Authors · 2024-11-17
> **Response (2/4)**
>
> **Q3:** It is unclear how Bad-PFL can address the limitations mentioned in Section 2.2. Though the authors mentioned using features of the target label as the trigger for a more effective backdoor attack, the benign mask $\delta$ is learned as a noise mask instead of the true features in the images. Could the authors argue this point?
>
> **Response:**
> Please refer to the response to Q1 for Reviewer XeVe.
>
>
> ---
>
> **Response:**
>
> **Q4:** The using of two masks now is not totally convincing and lacks of theoretical analysis to support this mechanism. What is exactly the role of each mask in the overall method and what is the relationship of these two masks? The paper may not fully justify why two separate masks are essential rather than using a single mask that can adaptively balance target and non-target features. How can we ensure that the two masks—the target feature enhancement mask ($\delta$) and the disruptive noise mask ($\xi$)—do not interfere with each other, either by collapsing into a single effect or by unintentionally complementing each other? Table 4 is not enough for this point.
>
> **Response:**
> **The relationship between the two masks.** Models classify samples based on their features. For a model to categorize a sample into the target label, the sample must contain features associated with the target label. Additionally, it is essential to disrupt the features that correspond to the sample's true label; otherwise, the model may still classify the sample based on the features associated with the true label. Therefore, both target feature and disruptive noise are crucial.
>
> **Clarification on using two masks.** While it is indeed possible to use a single adaptive mask, employing two masks effectively decouples the process, offering several advantages. First, it allows us to monitor whether the generator learns target features. Second, this is particularly advantageous in specific attack scenarios. For instance, all-to-all require attackers to classify data from one label to another. Using a single mask in a 10-class all-to-all attack means that each class must be targeted by a separate generator for every possible label pairing. This results in 90 unique combinations (10 classes × 9 other classes), which not only escalates attack costs but could also negatively impact the model's performance, as too many backdoor tasks could interfere with the primary task's learning. When employing two masks, we only need to train 10 generators, significantly improving efficiency. In fact, the benefits of decoupling in deep learning have been well established, and we won't elaborate further on that here. If you are interested, a quick Google search will yield numerous relevant cases and papers.
>
> **The learning of the two masks.**
> Regarding your concerns about potential interference between the two masks, please refer closely to Equation 7. In this equation, the generator is trained to adaptively produce the appropriate $\delta$ for $\xi$. $\xi$, derived from Equation 6, is designed to be disruptive, as it reduces the probability of the model classifying $x$ as its true label. Consequently, $x+\xi$ can be considered a sample with fewer features. Given this, looking back at Equation 7, to classify $x+\delta+\xi$ into the target label, the generator is compelled to produce features corresponding to the target label.
>
>
> ---
>
> **Q5:** In the threat model (Line 236-237), the attack can be colluded or non-colluded but the paper did not explicitly discuss or show this attack can work with both cases.
>
> **Response:** This manuscript assumes that the adversary can control a certain number of clients, leading to a collusive attack where these compromised clients work together. We also examine scenarios where the adversary is limited to controlling only a single client (Figure 2 in the original manuscript), in which case our attack can be considered non-collusive. Consequently, this manuscript covers both collusive and non-collusive situations.
>
> ---
>
> **Q6:** Can the authors explain more detail about the phenomenon mentioned in L263-264, and why the model should focus on the backdoor mask?
>
> **Response:** Models classify data based on the features present within these data. When we introduce target feature $\delta$ to a sample, the model utilizes both the target feature and the features associated with the true class for prediction. Disruptive noise $\xi$ serves to corrupt the features associated with the true class, thereby increasing the relative prominence of the target feature $\delta$. In this way, the model will place a greater focus on $\delta$ in its predictions.

---

> ### Author Response · Authors · 2024-11-17
> **Response (3/4)**
>
> **Q7:** Could the authors discuss the computational overhead of this method? Will it raise suspicion when it takes too long for the malicious to complete a training iteration?
>
> **Response:** Appendix C in the original manuscript discusses the computational overhead associated with our attack method. Specifically, we conduct simulation tests using an NVIDIA 4090 GPU and the CIFAR-10 dataset. On average, a benign client requires approximately 0.35 seconds to perform local training, while a malicious client takes about 0.77 seconds. At first glance, this may not seem significant enough to raise suspicion from the server. In fact, in federated learning, the main computational bottleneck lies in the communication costs between clients and the server. As a result, most backdoor attacks tend to focus more on performance rather than computational expense.
>
> ---
>
> **Q8:** In Line 294, what is the intuition of setting $\eta$ by the reversed sign of the input? Why does it ensure to return of an approximate solution?
>
> **Response:** Let us first recall Equation 6: $\xi = \max \ [\mathcal{L}(F(x + \xi;\theta_g), y)]$ where $||\xi|| \leq \sigma$. Here, $\sigma$ is assumed to be a small constant to ensure the imperceptibility of $\xi$ to humans. Then, we can perform a Taylor expansion of the optimization objective in Equation 6, yielding: $\mathcal{L}(F(x;\theta_g), y) +  \nabla_x^T \mathcal{L}(F(x;\theta_g), y) \xi.$ To maximize the objective function, it is essential to align the direction of $\xi$ with that of the gradient. Moreover, notice that $||\xi|| \leq \sigma$ (infinity norm) indicates that each element of $\xi$ is constrained between $-\sigma$ and $\sigma$, according to the definition of the infinity norm. Consequently, the analytical solution to Equation 6 is given by $\sigma \cdot sign (\nabla_{x} \mathcal{L}(F(x;\theta_g), y))$, where $sign(\cdot)$.
>
> ---
>
> **Q9:** Will another factor such as the number of clients affect the performance of this method, since it will affect the global model in each training round?
>
> **Response:** We have evaluated the impact of the number of clients on the performance of our attack method, with the results reported in the below table. Wherein, we fix the number of malicious clients at 10. We observe that as the total number of clients increases, both accuracy and ASR gradually decline. Though this, Bad-PFL still achieves significant ASRs (> 80%).
>
> | Client Number |  Acc  |  ASR  |
> |:-------------:|:-----:|:-----:|
> |       50      | 80.72 | 99.22 |
> |      100      | 80.29 | 97.95 |
> |      150      | 79.75 | 93.66 |
> |      200      | 78.04 | 85.87 |
>
> ---
>
> **Q10:** Some close related papers are missing such as [3, 4]. The authors can discuss the key difference in terms of the methodology of this work.
>
> **Response:** BapFL [3] primarily focuses on backdoor attacks targeting FedRep. Specifically, during the local training of malicious clients, [3] introduces random noise into classification heads to reduce their sensitivity to triggers. In contrast, this manuscript explores a wider spectrum of personalized federated learning methods and develops a fundamentally different technical approach. Besides, as data heterogeneity among clients increases, the divergence of optimal classification heads over different clients also grows, potentially diminishing their attack performance. In contrast, our method exhibits greater adaptability to higher levels of data heterogeneity.
>
> PFedBA [4] optimizes the triggers by aligning the gradients of samples with triggers to those without. The authors [4] explained that the similarity in gradients can partially represent the similarity in the model's decision boundaries. Compared to PFedBA, this manuscript delves deeper into why existing backdoor attack methods struggle with PFL, providing a comprehensive summary of the underlying reasons. Moreover, our attack method differs significantly from the technical approach of [4]. Our attack method offers higher stealthiness, as our triggers are sample-specific and imperceptible to humans, making them more challenging to detect (See Appendix B.2.11 for evaluation).

---

> ### Author Response · Authors · 2024-11-17
> **Response (4/4)**
>
> **Q11:** Can the authors elaborate more on the sharp difference in ASR among different datasets such as CIFAR100 and SVHN under FedREP in Tables 15 and 16?
>
> **Response:** We apologize for the data entry errors regarding the FedRep experimental results in Table 15. We have made corrections. We observe that existing backdoor attack methods often achieve significantly lower ASRs when applied to FedRep compared to other PFL methods. In FedRep, different clients share the same feature extractor, while their classification heads are trained separately on their respective datasets. This implies that, even if the feature extractor learns trigger patterns, the classification heads struggle to learn the mapping between the triggers and the target label due to the absence of trigger-containing data in the clean clients' datasets. Moreover, some studies [5] observed similar results.
>
> ---
>
>
> Reference:
>
> [1] Perdoor: Persistent non-uniform backdoors in federated learning using adversarial perturbations
>
> [2] Iba: Towards irreversible backdoor attacks in federated learning
>
> [3] Bapfl: You can backdoor personalized federated learning
>
> [4] Lurking in the shadows: Unveiling stealthy backdoor attacks against personalized federated learning
>
> [5] Revisiting personalized federated learning: Robustness against backdoor attacks
>
> [6] Backdoor defense with machine unlearning
>
> [7] Attack of the tails: Yes, you really can backdoor federated learning.
>
> [8] https://medium.com/@black_51980/novelty-in-science-8f1fd1a0a143

---

> ### Author Response · Authors · 2024-11-19
> **Discussion Inquiry**
>
> Dear Reviewer mG9G,
>
> We thank you for the precious review time and valuable comments. We have provided responses to your questions and the weakness you mentioned. We hope this can address your concerns.
>
> We would appreciate the opportunity to discuss whether your concerns have been addressed appropriately. Please let us know if you have any further questions or comments. We look forward to hearing from you soon.
>
> Best regards,
>
> Authors

---

> > ### Author Response · Authors · 2024-11-20
> > **Looking forward to your feedback**
> >
> > Dear Reviewer mG9G,
> >
> > Sorry to bother you again. We would like to know whether the responses have addressed your concerns.
> >
> > Should this be the case, we are encouraged that you raise the final rating to reflect this.
> >
> > If there are any remaining concerns, please let us know. We are more than willing to engage in further discussion and address any remaining concerns to the best of our abilities.
> >
> > We are looking forward to your reply. Thank you for your efforts in this manuscript.
> >
> > Best regards,
> >
> > Authors

---

> > > ### Comment · Reviewer_mG9G · 2024-11-20
> > >
> > > Thank you to the authors for the detailed responses, revisions, and extensive experiments. However, some of my concerns remain unresolved completely, and I would appreciate further discussion on the following points:
> > >
> > > - Q3: Could the authors provide a concise summary that directly addresses my original question? Specifically, how does this method tackle the three limitations outlined in Section 2.2? A point-by-point explanation would be helpful.
> > > - Q4: While I understand the intuition behind the two masks, $\xi$ and $\delta$, and their effects on the samples as illustrated in Figures 14 and 15, the relationship between these two masks remains unclear. For instance, are there cases where certain points in $\delta$ merely replace corresponding points in the learned mask $\xi$? Understanding the differences between the two masks (e.g., in terms of magnitude, overlap percentage, etc.) would provide better insights into their mechanism. This aspect has not been thoroughly addressed.
> > > - Q7: The computational overhead is provided in seconds but lacks a baseline comparison (e.g., performance without an attack). Without such a comparison, it is difficult to assess whether the overhead is insignificant. Could the authors provide a comparison against the baseline (no attack) or other attack strategies?
> > > - Q9: Is the observed performance degradation attributable to the setting itself (e.g., a larger number of clients inherently leading to poorer performance) or to the attack? A discussion on this distinction or experimental results would be valuable.
> > >
> > > I just leave my score unchanged for now.

---

> > > > ### Author Response · Authors · 2024-11-21
> > > > **Response to New Questions (1/3)**
> > > >
> > > > Thank you for your timely feedback and insightful questions. We appreciate the opportunity to engage in the discussion with you. According to your questions, we have made the corresponding revisions. The updates for Q4 can be found in Appendix B.2.4 and Appendix E, while the updates for Q5 and Q6 are detailed in Appendix C and Appendix B.2.7. Below, we offer a thorough response to each of your questions. If you have any concerns about our response or further questions, please let us know.
> > > >
> > > > ---
> > > >
> > > > **Q3:** Could the authors provide a concise summary that directly addresses my original question? Specifically, how does this method tackle the three limitations outlined in Section 2.2? A point-by-point explanation would be helpful.
> > > >
> > > > **Response:**
> > > > Our trigger consists of target features and disruptive noise. Data from the target class inherently contains these features and their relationship to the target label. Since models are trained to maximize accuracy, they will leverage any available features. Thus, as long as clients’ datasets include data from the target label, personalized models will inevitably learn these features and their connections.
> > > >
> > > > In full model-sharing methods, relying solely on the regularization term is insufficient, meaning the model fails to learn trigger patterns and the relationship between the trigger patterns and the target label. One intuitive approach is to inject poisoned data into the clients' datasets. Our attack method uses the natural features of the target class as triggers, which are present in clients' datasets.
> > > >
> > > > Similarly, in partial model-sharing methods, the challenge is conveying the connection between triggers and the target label to personalized models. Since we cannot alter the local training processes of benign clients, it is nearly impossible to embed the relationship between handcrafted triggers and the target label through data poisoning or other means. Instead, our attack method utilizes the natural features of the target label. This mapping between natural features and the target label, which already exists in clients' local datasets, allows us to effectively address the second challenge.
> > > >
> > > > Regarding the dilution of backdoors, we recognize that the clients' datasets contain these natural features and their relationships with the target label. During the fine-tuning or training process, the model is less likely to forget these relationships because doing so would lead to a decline in performance. In other words, the presence of these natural features in the training data maintains the model’s memory of the backdoor, mitigating the risk of the backdoor being overwritten or lost.
> > > >
> > > > ---
> > > >
> > > > **Q4:** While I understand the intuition behind the two masks, $\delta$ and $\xi$, and their effects on the samples as illustrated in Figures 14 and 15, the relationship between these two masks remains unclear. For instance, are there cases where certain points in $\delta$ merely replace corresponding points in the learned mask $\xi$? Understanding the differences between the two masks (e.g., in terms of magnitude, overlap percentage, etc.) would provide better insights into their mechanism. This aspect has not been thoroughly addressed.
> > > >
> > > >
> > > > **Response:** Thanks for your insightful question. We have evaluated the proportion of pixels where $\delta$ and $\xi$ share the same sign, finding it to be approximately 26.28\%, averaged over 1000 samples.
> > > > This indicates that $\delta$ and $\xi$ do not completely align in terms of the direction of pixel changes, suggesting a more intricate interplay between $\delta$ and $\xi$.
> > > >
> > > > In particular, Table 18 (Appendix B.2.4) reports the ASRs of our attack method when either $\delta$ or $\xi$ is fixed while varying the magnitude of the other.
> > > > We observe that our attack method is more sensitive to changes in $\epsilon$ (the magnitude of $\delta$), as $\delta$ represents the features of the target class.
> > > > In contrast, our attack method is relatively less sensitive to variations in $\sigma$ (the magnitude of $\xi$), since $\xi$ primarily serves to induce misclassification rather than explicitly directing the sample towards a specific class.
> > > > Nonetheless, $\xi$ remains essential; as $\sigma$ decreases, the performance of our attack method also gradually diminishes, although not as dramatically as when $\epsilon$ decreases.

---

> > > > > ### Author Response · Authors · 2024-11-21
> > > > > **Response to New Questions (2/3)**
> > > > >
> > > > > To further clarify the relationship between the $\delta$ and $\xi$, we have included visualizations of $\xi$ and $\delta+\xi$ to better illustrate their effects on pixel value changes (see Appendix E for visualizations).
> > > > > As illustrated in Figure 16, the pixel changes introduced by $\xi$ appear somewhat erratic from a human perspective.
> > > > > In contrast, the combined effect of $\xi+\delta$ exhibits a clear pattern, predominantly altering pixels in the upper right corner.
> > > > > This highlights the interplay between $\delta$ and $\xi$, characterized by both resistance and agreement.
> > > > > While $\xi$ proposes specific pixel change directions, $\delta$ can either amplify or counteract these suggestions.
> > > > > This means that $\delta+\xi$ reflects a negotiation between the two: $\delta$ may dampen or redirect some of the changes suggested by $\xi$.
> > > > > This dynamic can lead to concentrated perturbations in certain areas of the input, indicating that $\xi$ selectively agrees with the changes proposed by $\delta$.
> > > > > This phenomenon can be observed in Figures 17, 18, and 19, reinforcing the notion that there exists a complex interaction between $\xi$ and $\delta$, rather than a straightforward combination into a single effect.
> > > > >
> > > > > ---
> > > > >
> > > > > **Q5:** The computational overhead is provided in seconds but lacks a baseline comparison (e.g., performance without an attack). Without such a comparison, it is difficult to assess whether the overhead is insignificant. Could the authors provide a comparison against the baseline (no attack) or other attack strategies?
> > > > >
> > > > > **Response:** We here discuss the overhead associated with our attack method, examining both the training and inference phases.
> > > > > During the FL process, our attack method involves the optimization of the generator and the training of the global model on trigger-added data.
> > > > > On the one hand, the optimization of the generator, as described in Equation 7, requires two complete forward and backward passes of the global model, along with one forward and backward pass of the generator.
> > > > > On the other hand, optimizing the global model on trigger-added data involves crafting triggers, which entails a single forward pass of the generator (for $\delta$), as well as a forward and backward pass of the global model (for $\xi$).
> > > > >
> > > > > The below table reports the empirically time (in seconds) required for compromised clients to execute local training using various attack methods.
> > > > > We conduct these experiments using CIFAR-10, with the reported times averaged over 100 trials on a single RTX 4090 GPU.
> > > > > "No Attack" indicates the time taken for a client to perform local training without executing backdoor attacks.
> > > > > The below table does not report the costs associated with LF-Attack, as it needs training models from scratch multiple times (in a linear relationship with the number of layers in the neural networks) to evaluate each layer's significance for backdoor attacks.
> > > > > The attack costs for LF-Attack are significantly higher than those of existing backdoor attack methods, and we will not discuss it further.
> > > > >
> > > > > We observe that Neurotoxin incurs the lowest attack overhead since it utilizes a fixed trigger; however, this also results in lower attack performance.
> > > > > More advanced backdoor attack methods often employ more sophisticated trigger generation techniques.
> > > > > For instance, Perdoor uses the BIM method to create triggers, necessitating multiple complete forward and backward passes of the global model (10 times here).
> > > > > PFedBA has to handle a gradient matching problem, requiring at least two forward and backward passes of the global model for each optimization iteration of the trigger.
> > > > > Our attack method also demands a certain amount of time investment.
> > > > > Nevertheless, we stress that compared to existing attack methods, our attack method still achieves superior performance while maintaining a competitive time overhead.
> > > > > Moreover, federated backdoor attack methods focus more on attack performance over runtime costs, as the primary bottleneck in FL lies in communication costs.
> > > > > These attack methods usually require only a few seconds, which is small compared to communication durations, making them less detectable in practice.
> > > > > In the inference phase, our method for generating triggers for 32 data samples takes approximately 0.07 seconds, which is also quite efficient.
> > > > > In summary, our attack method is practical.
> > > > >
> > > > > |   Attack   | FedProx | SCAFFOLD | FedBN | FedRep | Ditto |
> > > > > |:----------:|:-------:|:--------:|:-----:|:------:|:-----:|
> > > > > |  No Attack |  0.453  |   0.211  | 0.201 |  0.447 | 0.451 |
> > > > > | Neurotoxin |  0.475  |   0.223  | 0.213 |  0.452 | 0.468 |
> > > > > |   Perdoor  |  5.744  |   3.273  | 3.113 |  3.349 | 3.358 |
> > > > > |     Iba    |  0.791  |   0.661  | 0.620 |  1.227 | 1.178 |
> > > > > |    BapFL   |  0.982  |   0.578  | 0.552 |  0.797 | 0.552 |
> > > > > |   PFedBA   |  1.820  |   1.540  | 1.480 |  1.649 | 1.443 |
> > > > > |    Ours    |  0.818  |   0.620  | 0.613 |  1.206 | 1.132 |

---

> > > > > > ### Author Response · Authors · 2024-11-21
> > > > > > **Response to New Questions (3/3)**
> > > > > >
> > > > > > **Q6:** Is the observed performance degradation attributable to the setting itself (e.g., a larger number of clients inherently leading to poorer performance) or to the attack? A discussion on this distinction or experimental results would be valuable.
> > > > > >
> > > > > > **Response:**
> > > > > > This is an interesting question.
> > > > > > Specifically, we conduct backdoor attacks with a fixed number of compromised clients.
> > > > > > As the number of clients increases, the expected time for compromised clients to be selected will naturally extend, leading to a decline in attack performance.
> > > > > > In extreme cases, when the number of clients approaches infinity, the probability of compromised clients participating in FL process becomes negligible.
> > > > > >
> > > > > > We have conducted experiments with a fixed ratio of compromised clients (10%), and the results are shown in the table below.
> > > > > > In this setting, we see that the ASRs of our attack method returns to nearly 100%.
> > > > > > Therefore, we think that the observed performance decline is primarily due to the setting.
> > > > > >
> > > > > > |    Strategy   | Fixed Number |       | Fixed Ratio |       |
> > > > > > |:-------------:|:------------:|:-----:|:-----------:|:-----:|
> > > > > > | Client Number |      Acc     |  ASR  |     Acc     |  ASR  |
> > > > > > |       50      |     80.72    | 99.22 |    80.65    | 98.00 |
> > > > > > |      100      |     80.29    | 97.95 |    80.24    | 97.68 |
> > > > > > |      150      |     79.75    | 93.66 |    79.47    | 99.30 |
> > > > > > |      200      |     78.04    | 85.87 |    77.95    | 98.85 |

---

> > > > > > > ### Author Response · Authors · 2024-11-22
> > > > > > > **Looking forward to your feedback**
> > > > > > >
> > > > > > > Dear Reviewer mG9G,
> > > > > > >
> > > > > > > We have shared the responses to the latest questions you raised. We would like to know if the responses can address your concerns. Thank you for your time and effort.
> > > > > > >
> > > > > > > Best regards,
> > > > > > >
> > > > > > > Authors

---

> > > > > > > > ### Author Response · Authors · 2024-11-24
> > > > > > > >
> > > > > > > > Dear Reviewer mG9G,
> > > > > > > >
> > > > > > > > Sorry to bother you again. As the discussion phase is wrapping up, we would appreciate it if you could take some time to read our response to your latest questions.
> > > > > > > >
> > > > > > > > If you feel that the response addresses your concerns, we would be thankful if you could consider raising the final rating. If not, we would love to hear any further comments. Please let us know.
> > > > > > > >
> > > > > > > > Thank you once again for your support and understanding.
> > > > > > > >
> > > > > > > > Best regards,
> > > > > > > >
> > > > > > > > Authors

---

> > > > > > > > > ### Author Response · Authors · 2024-11-25
> > > > > > > > >
> > > > > > > > > Dear Reviewer mG9G,
> > > > > > > > >
> > > > > > > > > As we approach the final day of the discussion period, we would like to know if our latest responses have addressed your concerns. If you have any remaining questions or require further clarification, we would be more than happy to provide additional details. Your support would mean a great deal to us and would greatly encourage our continued efforts in this area.
> > > > > > > > >
> > > > > > > > > Thank you once again for your time, effort, and constructive comments!
> > > > > > > > >
> > > > > > > > > Best regards,
> > > > > > > > >
> > > > > > > > > Authors

---

> ### Author Response · Authors · 2024-11-25
>
> Dear Reviewer mG9G,
>
> We are greatly encouraged to see that Reviewer XeVe has indicated all his concerns have been addressed and made a positive final rating. We notice that your question Q3 is also one of the concerns raised by Reviewer XeVe, who suggested that this concern has been addressed. We hope that our responses can also address your concern regarding Q3. In response to your comments on Q4, Q7, and Q9, we have added the corresponding experiments and discussions according to your suggestions. We would greatly appreciate it if you could take a moment to read our responses. We look forward to hearing from you.
>
> Best regards,
>
> Authors

---

> ### Comment · Reviewer_mG9G · 2024-11-25
>
> Thanks to the authors for their hard work and detailed responses. The explanations and clarifications addressed most of my concerns. As a result, I have decided to increase my score to support this paper. I recommend incorporating these detailed responses into the paper to help readers better understand the work and provide opensource for reproducibility.

---

> > ### Author Response · Authors · 2024-11-26
> >
> > Many thanks for your feedback. We appreciate your recognition. We have included these detailed discussions in the revised paper. Our codes are included in the supplementary materials, and we will also make the codes publicly available.

---

### Official Review · Reviewer_XeVe · 2024-11-04

**Soundness:** 3
**Presentation:** 4
**Contribution:** 3
**Rating:** 6
**Confidence:** 4

**Summary:**

In this paper, the authors propose a novel PFL backdoor method that leverages natural features from the data as triggers, rather than manually designed triggers used in previous attacks. Specifically, the proposed method adopt a generator to generate the features that make samples appear similar to the target category and introduces disruptive noise to eliminate features associated with the ground-truth labels.

**Strengths:**

1. The proposed method is novel in that the trigger is designed to be sample-specific, which is a significant difference with the previous works

2. The paper is well organized and is easy to follow

3. The authors provide a thorough discussion about the challenges of backdoor attack under the PFL setting.

4. The authors conduct the experiment on three benchmark datasets. Besides, the effectiveness of the propsoed backdoor methods are also evaluated under the state-of-the-art defense mechanisms

**Weaknesses:**

1. In this paper, the authors lack the in-depth discussion about why the propose method can overcome the challenges metioned in section 2.2.

	a. It is suggested that the authors give more intuitive or theoretical discussion about why it works.

	b. Specifically, the authors may give more experimental analysis about the inherent mechanism about the proposed method. For example, the authors can visualize the representation of different classes under clean and backdoored data.


2. Some related attacks are missing, such as:
	a. Lurking in the shadows: Unveiling Stealthy Backdoor Attacks against Personalized Federated Learning (https://arxiv.org/html/2406.06207v1)


3. Missing defense method.
	a. Simple-Tuning: Clients reinitialize their classifiers and then retrain them using their local clean datasets while keeping the feature encoder fixed. (https://dl.acm.org/doi/10.1145/3580305.3599898)

**Questions:**

See weekness 1, 2, 3

---

> ### Author Response · Authors · 2024-11-17
> **Response (1/3)**
>
> We appreciate the time and effort you invested in reviewing this manuscript. Your insightful comments greatly enhance the depth and quality of this manuscript. We have uploaded the revised manuscript, with changes highlighted in red for your convenience. Specifically, the revision addressing Q1 can be found in Appendix D, while those related to Q2 \& Q3 are located in Appendix B.2.10. Below, we provide a detailed response to each of your comments. Unless otherwise specified, the experimental configuration defaults to that described in Section 4.2.
>
> ---
>
> **Q1:** The authors lack the in-depth discussion about why the propose method can overcome the challenges metioned in section 2.2.
>
> **Response:**
> We now explain how our attack method effectively overcomes the above three challenges. The trigger employed in our attack method consists of target feature ($\delta$) and disruptive noise ($\xi$). Naturally, data from the target class inherently contains $\delta$ and the relationship between $\delta$ and the target label (established through human labeling). Recall that we train models to maximize accuracy. Consequently, models tend to leverage any available features to do so. This means that as long as the clients' datasets include data from the target label, personalized models will inevitably learn $\delta$ and the relationship between $\delta$ and the target label. This enables our method to effectively address the aforementioned challenges.
>
> More specifically, in full model-sharing methods, relying solely on the regularization term is inadequate for transferring the backdoor to personalized models. Our attack method leverages the natural features of the target class as our trigger, which are inherently present in the data associated with that class, including the local datasets of benign clients. Personalized models trained on benign clients' local datasets will actively learn the natural features and the relationship from the natural features to the target label for higher accuracy. The guidance provided by the regularization term also further enhances this learning process, allowing our attack method to effectively overcome the first challenge.
>
> In partial model-sharing methods, the challenge lies in effectively conveying the connection between the triggers and the target label to the personalized models. Since we cannot alter the local training processes of benign clients, it is nearly impossible to embed the relationship between handcrafted triggers and the target label through data poisoning or other means. Instead, our attack method utilizes the natural features of the target label. This mapping between natural features and the target label, which already exists in the local datasets of benign clients, allows us to effectively address the second challenge without needing to modify the training process of benign clients.
>
> Regarding the dilution of backdoors, we recognize that the clients' datasets contain these natural features and their relationships with the target label. During the fine-tuning or training process, the model is less likely to forget these relationships because doing so would lead to a decline in performance. In other words, the presence of these natural features in the training data reinforces the model’s memory of the backdoor, mitigating the risk of it being overwritten or lost. In summary, the above analysis clearly illustrates how our attack method successfully overcomes the three challenges previously mentioned.
>
> Furthermore, it is important to note that even if a particular client’s dataset does not include data from the target label, the effectiveness of our attack method is likely to persist. First, in practice, only a small number of client datasets may lack data from the target class, making it unlikely that the global model fails to learn $\delta$ and the mapping from $\delta$ to the target label. Moreover, in our attack method, malicious clients actively promote the model's reliance on $\delta$ to predict the target class, as indicated in Equation 7. The similarity constraint between the global model and the personalized models encourages the personalized models to leverage the relationship between $\delta$ and the target class more effectively. This encourages the personalized models to also utilize the relationship between $\delta$ and the target class to a greater extent. Second, we introduce destructive noise $\xi$, which interferes with features belonging to the true class, thereby allowing $\delta$ to function more effectively in the decision-making process of personalized models. These two unique designs can enhance the performance of our attack method. By the way, the only potential countermeasure we can conceive against our attack would be if clients fine-tune their personalized models without including data from the target class. However, the absence of target class data would significantly impair the performance of these personalized models on the target class.

---

> ### Author Response · Authors · 2024-11-17
> **Response (2/3)**
>
> **Empirical evidence:** To further substantiate our claims, we present experimental results. First, we demonstrate that $\delta$ utilized in our attack method are indeed natural features of the target class. We employ T-SNE to visualize the features extracted from the test data by the global model. We also generate $\delta$ for these test samples and visualize the features extracted from $\delta$. As illustrated in Figure 14 (see revised manuscript), the model classifies $\delta$ as belonging to the target class, indicating that it recognizes $\delta$ as natural features of the target class.
>
> Next, we validate the effectiveness of the disruptive noise $\xi$. Similarly, we use T-SNE to visualize the features of both the test samples with and without $\xi$. Figure 15 (see revised manuscript) reveals that, while the features from $x$ cluster neatly by class, those from $x + \xi$ exhibit a more chaotic distribution. This confirms that $\xi$ effectively disrupts the features associated with their ground-truth classes.
>
> We also conduct numerical experiments to further substantiate our conclusions. We train a ResNet10 on the CIFAR-10 dataset from scratch using three distinct configurations. The first configuration is a standard training setup. In the second configuration, we add disruptive noise (Equation 6 in the original paper) to the training samples of the target label during each iteration. Building on the second configuration, the third configuration introduces $\delta$ into the training samples of the target label. Intuitively, the disruptive noise is expected to corrupt the features of the training samples of the target label, which would hinder the model to learn the underlying features of the target label, resulting in poor performance on those samples. In the third configuration, if our $\delta$ accurately captures the features of the target label, we anticipate that the model will learn more about the target label compared to the second configuration, leading to better performance in the samples of target label. We reuse the generator in Section 4.2 of the original manuscript (against FedRep).
>
> The below table reports the accuracy of the model on the entire test set of CIFAR-10, as well as on the test samples from the target label alone. We observe that the model achieves an accuracy of only 6.70% on the samples of the target label, indicating that the disruptive noise indeed significantly impairs the features of the samples of the target label. In the third configuration, we see that the model's accuracy on the samples of the target label rebounds to 39.1%. This suggests that our generator learns the features of the target label.
>
> |     Setup     |  Acc  | Acc of the Target Label |
> |:-------------:|:-----:|:-----------------------:|
> |  First Config (Standard Training) |  80.7 |           80.3          |
> | Second Config (with $\xi$) |  70.7 |           6.7           |
> |  Third Config (with $\delta+\xi$) |  72.9 |           39.1          |

---

> ### Author Response · Authors · 2024-11-17
> **Response (3/3)**
>
> **Q2:** Some related attacks are missing: Lurking in the shadows: Unveiling Stealthy Backdoor Attacks against Personalized Federated Learning (https://arxiv.org/html/2406.06207v1).
>
> **Response:**
> In response to your suggestion, we have incorporated the works [4,5] you mentioned, with specific results reported in the table below. We employ FedRep, and this evaluation has been included in the revised paper (Appendix B.2.10).
> In line with suggestions from other reviewers, we have also added two backdoor defenses, BAERASER [6] and MAD [7], and three backdoor attacks, Perdoor [1], Iba [2], and BapFL [3].
> Overall, Bad-PFL shows a significant advantage over these attacks in terms of ASRs when evaluated against three different defenses.
> More detailed discussions can be found in Appendix B.2.10.
>
>
> |   Defense  | Simple-Tuning [5] |        | BAERASER [6] |        |   MAD [7]  |        |
> |:----------:|:-------------:|:------:|:--------:|:------:|:------:|:------:|
> |   Attack   |      Acc      |   ASR  |    Acc   |   ASR  |   Acc  |   ASR  |
> | Neurotoxin |     82.65     | 19.80  |  78.59   | 13.05  | 74.52  | 19.46  |
> |  LF-Attack |     81.68     | 12.59  |  77.90   | 15.24  | 74.81  | 10.49  |
> |   Perdoor [1]  |     81.59     | 63.15  |  79.33   | 84.12  | 74.64  | 46.90  |
> |     Iba [2]     |     81.82     | 49.31  |  77.74   | 78.98  | 74.55  | 55.58  |
> |    BapFL [3]   |     82.24     | 22.79  |  79.39   | 17.59  | 74.75  | 24.73  |
> |   PFedBA [4]   |     81.27     | 42.36  |  78.59   | 31.88  | 74.29  | 55.92  |
> |     Our    |     82.05     | 88.82  |  77.68   | 91.54  | 74.37  | 90.74  |
>
> ---
>
>
> **Q3:** Missing defense method: Simple-Tuning: Clients reinitialize their classifiers and then retrain them using their local clean datasets while keeping the feature encoder fixed. (https://dl.acm.org/doi/10.1145/3580305.3599898).
>
> **Response:** See Q2.
>
> ---
>
>
> [1] Perdoor: Persistent non-uniform backdoors in federated learning using adversarial perturbations
>
> [2] Iba: Towards irreversible backdoor attacks in federated learning
>
> [3] Bapfl: You can backdoor personalized federated learning
>
> [4] Lurking in the shadows: Unveiling stealthy backdoor attacks against personalized federated learning
>
> [5] Revisiting personalized federated learning: Robustness against backdoor attacks
>
> [6] Backdoor defense with machine unlearning
>
> [7] Multi-metrics adaptively identifies backdoors in federated learning
>
> [8] Neural cleanse: Identifying and mitigating backdoor attacks in neural networks
>
> [9] Strip: A defence against trojan attacks on deep neural networks

---

> > ### Comment · Reviewer_XeVe · 2024-11-25
> >
> > Thanks for authors' feedback! The authors provide the discussion about why the proposed method work. Also the missing defense methods are included. I would like to maintain my score towards acceptance.

---

> > > ### Author Response · Authors · 2024-11-25
> > >
> > > We are delighted to hear that all your concerns have been addressed. We also appreciate your recognition and support.

---

> ### Author Response · Authors · 2024-11-19
> **Discussion Inquiry**
>
> Dear Reviewer XeVe,
>
> We thank you for the precious review time and valuable comments. We have provided responses to your questions and the weakness you mentioned. We hope this can address your concerns.
>
> We would appreciate the opportunity to discuss whether your concerns have been addressed appropriately. Please let us know if you have any further questions or comments. We look forward to hearing from you soon.
>
> Best regards,
>
> Authors

---

> > ### Author Response · Authors · 2024-11-22
> > **Looking forward to your feedback**
> >
> > Dear Reviewer XeVe,
> >
> > Sorry to bother you again. With the discussion phase nearing the end, we would like to know whether the responses have addressed your concerns.
> >
> > Should this be the case, we are encouraged that you raise the final rating to reflect this.
> >
> > If there are any remaining concerns, please let us know. We are more than willing to engage in further discussion and address any remaining concerns to the best of our abilities.
> >
> > We are looking forward to your reply. Thank you for your efforts in this manuscript.
> >
> > Best regards,
> >
> > Authors

---

### Official Review · Reviewer_ZRox · 2024-11-04

**Soundness:** 3
**Presentation:** 3
**Contribution:** 2
**Rating:** 6
**Confidence:** 3

**Summary:**

The authors develop Bad-PFL, a new backdoor attack that leverages natural features from the target label as a trigger, enabling backdoor persistence across both global and personalized models. It uses a dual-component trigger, combining natural target features and disruptive noise to deceive the model without being easily detectable. Bad-PFL demonstrates high attack success rates even when state-of-the-art defenses are applied.

**Strengths:**

It introduces Bad-PFL, a unique attack using natural features as triggers in PFL, and a novel dual-trigger design combining natural features and disruptive noise for stealth and effectiveness. Experiments are thorough, covering multiple datasets and comparing Bad-PFL to six top backdoor attacks, showing its high success rate even with strong defenses. This work highlights vulnerabilities in PFL previously thought resistant to backdoors, encouraging the development of specialized defenses and impacting fields that rely on PFL.

**Weaknesses:**

1) Some newer or adaptive defenses, such as gradient masking or input filtering, are not covered, leaving questions about Bad-PFL's effectiveness against more dynamic defenses.

2) While Bad-PFL claims high stealthiness, there is limited quantitative analysis on how detectable the trigger is by modern anomaly or intrusion detection systems.

3) The experiments are largely conducted on ResNet models, leaving questions about the attack's adaptability to other architectures commonly used in federated learning, such as transformer-based models.

**Questions:**

1) Have you tested the detectability of the Bad-PFL trigger with current anomaly or backdoor detection methods? If so, what results did you observe?

2) How does the attack’s success vary with different levels of data heterogeneity among clients? Could more heterogeneous data distributions reduce Bad-PFL’s attack success rate or persistence?

3) How does Bad-PFL perform across different model architectures, such as transformer-based models?

---

> ### Author Response · Authors · 2024-11-17
> **Response (1/2)**
>
> We appreciate the time and effort you dedicated to reviewing this manuscript. Your insightful comments significantly enrich this manuscript. We have uploaded the revised manuscript, and the changes are highlighted in red for your convenience. Below, we provide a detailed response to each of your comments. Unless otherwise stated, the experimental configuration defaults to that used in Section 4.2.
>
> ---
>
> **Q1:** Some newer or adaptive defenses, such as gradient masking or input filtering, are not covered, leaving questions about Bad-PFL's effectiveness against more dynamic defenses.
>
> **Response:**
> According to your suggestion, we have included the latest defense methods, with specific results reported in the table below. We employ FedRep, and this evaluation has been included in the revised paper (Appendix B.2.10). Notably, Multi-metrics Adaptive Defense (MAD) represents a state-of-the-art dynamic adaptive backdoor defense. Additionally, Simple-Tuning and BAERASER are two state-of-the-art post-hoc defense methods. BAERASER employs forgetting techniques to eliminate the model's memory of triggers. Furthermore, based on suggestions from other reviewers, we have also added four advanced backdoor attack methods: Perdoor, Iba, BapFL, and PFedBA. Overall, Bad-PFL demonstrates a significant advantage over these attacks in terms of ASRs against three defenses. More detailed discussions can be found in Appendix B.2.10.
>
> |   Defense  | Simple-Tuning [5] |        | BAERASER [6] |        |   MAD [7]  |        |
> |:----------:|:-------------:|:------:|:--------:|:------:|:------:|:------:|
> |   Attack   |      Acc      |   ASR  |    Acc   |   ASR  |   Acc  |   ASR  |
> | Neurotoxin |     82.65     | 19.80  |  78.59   | 13.05  | 74.52  | 19.46  |
> |  LF-Attack |     81.68     | 12.59  |  77.90   | 15.24  | 74.81  | 10.49  |
> |   Perdoor [1]  |     81.59     | 63.15  |  79.33   | 84.12  | 74.64  | 46.90  |
> |     Iba [2]     |     81.82     | 49.31  |  77.74   | 78.98  | 74.55  | 55.58  |
> |    BapFL [3]   |     82.24     | 22.79  |  79.39   | 17.59  | 74.75  | 24.73  |
> |   PFedBA [4]   |     81.27     | 42.36  |  78.59   | 31.88  | 74.29  | 55.92  |
> |     Our    |     82.05     | 88.82  |  77.68   | 91.54  | 74.37  | 90.74  |
>
>
> ---
>
> **Q2:** While Bad-PFL claims high stealthiness, there is limited quantitative analysis on how detectable the trigger is by modern anomaly or intrusion detection systems.
>
> **Response:**
> We have included an evaluation of Bad-PFL' stealthiness (Appendix B.2.11) from two perspectives: 1) whether benign clients can detect backdoors in their models, and 2) whether benign clients can recognize trigger-added samples.
>
> For the first perspective, we utilize Neural Cleanse, which computes an anomaly index by recovering trigger candidates to convert all clean images to each label. If the anomaly index for a specific label is significantly higher than for others, it indicates that the model is likely compromised. We evaluate different attack methods by calculating the anomaly index for the target label using Neural Cleanse. A smaller anomaly index suggests that the backdoor attack is harder to detect. For the second perspective, we employ STRIP, which identifies trigger-added samples based on the prediction entropy of input samples generated by applying different image patterns. Higher entropy signifies a more stealthy trigger.
>
> We train ResNet10 with FedRep on CIFAR-10. By default, we select the models of the first ten benign clients and the CIFAR-10 test set to estimate the anomaly index and entropy. The below table reports the detection results. The average anomaly index for non-target labels is 1.9, while the entropy of clean samples is 0.92. We see that our attack method achieves a lower anomaly index and higher entropy compared to baseline attacks, demonstrating superior stealthiness.
>
> | Detection Method | Neural Cleanse (Anomaly Index) [8] | STRIP (Entropy) [9] |
> |:----------------:|:------------------------------:|:---------------:|
> |    Neurotoxin    |               5.8              |       0.13      |
> |     LF-Attack    |               5.7              |       0.12      |
> |      PFedBA      |               4.9              |       0.25      |
> |        Our       |               2.2              |       0.77      |

---

> ### Author Response · Authors · 2024-11-17
> **Response (2/2)**
>
> **Q3:** The experiments are largely conducted on ResNet models, leaving questions about the attack's adaptability to other architectures commonly used in federated learning, such as transformer-based models.
>
> **Response:**
> We have evaluated the performance of our attack in vision transformer (Appendix B.2.2).
>
> Specifically, we use a vision transformer (ViT) pre-trained on ImageNet (provided by Torchvision) as the initialization for the server, with the classification head reinitialized to accommodate CIFAR-10. We employ FedRep for this evaluation. The below table reports the attack results. As can be seen, our attack method performs well on ViT, achieving an ASR of 98.94%.
>
> |  ViT  | Neurotoxin | LF-Attack |  Our  |
> |:-----:|:----------:|:---------:|:-----:|
> |  Acc  |    84.43   |   85.06   | 84.89 |
> |  ASR  |    35.64   |   20.58   |  98.9 |
>
> ---
>
>
> **Q4:** How does the attack’s success vary with different levels of data heterogeneity among clients? Could more heterogeneous data distributions reduce Bad-PFL’s attack success rate or persistence?
>
> **Response:**
> Figure 5 in the original manuscript evaluates the attack performance of our attack method under varying degrees of data heterogeneity. As heterogeneity increases, the performance of all backdoor attack methods tends to decline, which appears to be inevitable. In detail, the higher levels of data heterogeneity suggest greater divergence between the optimal models of different clients. In this context, benign clients must distance their personalized models from the global model to ensure good performance on their own datasets, making it more challenging for compromised clients to conduct attacks.
>
>
> ---
>
> [1] Perdoor: Persistent non-uniform backdoors in federated learning using adversarial perturbations
>
> [2] Iba: Towards irreversible backdoor attacks in federated learning
>
> [3] Bapfl: You can backdoor personalized federated learning
>
> [4] Lurking in the shadows: Unveiling stealthy backdoor attacks against personalized federated learning
>
> [5] Revisiting personalized federated learning: Robustness against backdoor attacks
>
> [6] Backdoor defense with machine unlearning
>
> [7] Multi-metrics adaptively identifies backdoors in federated learning
>
> [8] Neural cleanse: Identifying and mitigating backdoor attacks in neural networks
>
> [9] Strip: A defence against trojan attacks on deep neural networks

---

> ### Author Response · Authors · 2024-11-19
> **Discussion Inquiry**
>
> Dear Reviewer ZRox,
>
> We thank you for the precious review time and valuable comments. We have provided responses to your questions and the weakness you mentioned. We hope this can address your concerns.
>
> We would appreciate the opportunity to discuss whether your concerns have been addressed appropriately. Please let us know if you have any further questions or comments. We look forward to hearing from you soon.
>
> Best regards,
>
> Authors

---

> > ### Author Response · Authors · 2024-11-22
> > **Looking forward to your feedback**
> >
> > Dear Reviewer ZRox,
> >
> > Sorry to bother you again. With the discussion phase nearing the end, we would like to know whether the responses have addressed your concerns.
> >
> > Should this be the case, we are encouraged that you raise the final rating to reflect this.
> >
> > If there are any remaining concerns, please let us know. We are more than willing to engage in further discussion and address any remaining concerns to the best of our abilities.
> >
> > We are looking forward to your reply. Thank you for your efforts in this manuscript.
> >
> > Best regards,
> >
> > Authors

---

### Meta-Review · Area_Chair_ybKk · 2024-12-18

**Metareview:**

The paper proposes a new backdoor attack on personalized federated learning. Specifically, it first generates natural features associated with the target label. Based on this natural feature set, it further finds a perturbation to strengthen the feature-label associations. The experiments show the proposed attack is effective against several pFL methods. All reviewers agree the paper is well-written and the findings are interesting. The experiments are also comprehensive and the results look promising. However, the reviewers also point out several concerns. For example, there is not enough discussion on the challenges in the backdoor with pFL and why the proposed method can overcome this challenge. Several ablations on the key model designs and potential defenses are missing. Please revise the paper based on the reviewer's suggestions.

**Additional Comments On Reviewer Discussion:**

The reviewer raised several concerns about the clarification, asking for some additional experiments and some suggestions on adding the discussion with the proposed method's strategy to solve the challenge. The author did a good job of addressing on reviewer's concerns by adding several supporting experiments and adding the discussion in the revision.

---

### Decision · Program_Chairs · 2025-01-22

Accept (Poster)